# Decoupled-Value Attention for Prior-Data Fitted Networks: GP Inference for Physical Equations

## Abstract

Prior-data fitted networks (PFNs) are a promising alternative to time-consuming Gaussian process (GP) inference for creating fast surrogates of physical systems. PFN reduces the computational burden of GP-training by replacing Bayesian inference in GP with a single forward pass of a learned prediction model. However, with standard Transformer attention, PFNs show limited effectiveness on high-dimensional regression tasks. We introduce Decoupled-Value Attention (DVA)–motivated by the GP property that the function space is fully characterized by the kernel over inputs and the predictive mean is a weighted sum of training targets. DVA computes similarities from inputs only and propagates labels solely through values. Thus, the proposed DVA mirrors the GP update while remaining kernel-free. We demonstrate that PFNs are backbone architecture invariant and the crucial factor for scaling PFNs is the attention rule rather than the architecture itself. Specifically, our results demonstrate that (a) localized attention consistently reduces out-of-sample validation loss in PFNs across different dimensional settings, with validation loss reduced by more than 50% in five- and ten-dimensional cases, and (b) the role of attention is more decisive than the choice of backbone architecture, showing that CNN, RNN and LSTM-based PFNs can perform at par with their Transformer-based counterparts. The proposed PFNs provide 64-dimensional power flow equation approximations with a mean absolute error of the order of $10^{-3}$, while being over $80\times$ faster than exact GP inference.

## 1 Introduction

Bayesian inference provides a powerful framework for reasoning under uncertainty, with methods like Gaussian processes (GPs) offering well-calibrated predictions and principled uncertainty estimates (Williams & Rasmussen, 2006). However, the practical application of these methods is often hindered by the heavy computational burden of learning kernel hyperparameters. For example, exact GP inference scales cubically with the number of data points, making its deployment infeasible for large datasets or problems requiring repeated training. Consider a physical system where a surrogate GP is chosen due to its uncertainty estimates and differentiable closed-form expressions. However, the underlying input dataset and configuration changes frequently, and the surrogate is supposed to work for these new, previously unseen variations. For example, changes in underlying physical networks for power grids Tan et al. (2025). In such conditions, GP needs to be trained repeatedly, incurring significant computing cost, each time the dataset changes.

To address this, Prior-Data Fitted Networks (PFNs) have emerged as a method (Müller et al., 2022) that uses large-scale pre-training to approximate the Bayesian posterior predictive in a single forward pass. Note that unlike sparse GP approximations Daskalakis et al. (2022), PFNs eliminate kernel parameter training step. Although Low-rank approximations reduce GP cost to $\mathcal{O}(nm^2)$, where $m$ is a user-defined parameter, PFNs need only a forward pass at deployment. This advantage grows when multiple GPs must be learned, as training $K$ GPs scales to $\mathcal{O}(Knm^2)$, with each $m$ requiring tuning. PFNs avoid these issues by directly predicting the posterior distribution in one step in a forward pass of the trained network. However, PFNs face scaling and bias issues in problems with high input dimensions due to their joint input–output embedding strategy Müller et al. (2022); Hollmann et al. (2025); Wang et al. (2025); Nagler (2023). Attention over concatenated $(\mathbf{x}, \mathbf{y})$ embeddings, as done in PFNs, degrade locality and similarity measures as input dimension grows

Further, they are almost exclusively built using Transformer architectures, which have high memory requirements. These challenges in existing PFFs motivate this work.

In this work, we propose **Decoupled-Value Attention (DVA)**, an input localized attention mechanism to scale PFNs with different architectures. We provide evidence that the attention mechanism is the primary driver of PFN performance, and it can be built using different architectures Convolution Neural Networks (CNNs) as well along with Transformers. The proposed DVA computes attention affinities (queries and keys) purely from the input space, while propagating information from the output space exclusively through the values. This aligns directly with the functional-space view of a GP, where the influence of training outputs $y_i$ on a test prediction is weighted by the similarity of their corresponding inputs $x_i$ Williams & Rasmussen (2006). This is a significant deviation from the standard attention mechanism applied in existing PFN works where affinities are calculated from a concatenated input-output vector Müller et al. (2022); Hollmann et al. (2022). This, combining inputs and outputs, increases the computational load, reducing PFNs ability to learn when the dimensions of input space grow. We note the observation made by Nagler (2023) that the convergence of PFNs is due to the attention mechanism, while bias[1] is a function of architecture choice. More importantly, it argues that a *post-hoc localization mechanism* is needed to reduce bias. We provide both theoretical and empirical evidence that the proposed localized attention weights are proportional to the query–context distance during PFN inference. Further, the reduction in PFN validation loss is consistent across architectures. Experimental studies show that DVA performs better than standard Vanilla Attention (VA) used in PFN literature, across dimensions and architectures. Our main contributions are:

- **A Localized Attention Mechanism for GP-PFNs:** We introduce DVA, which explicitly enforces input-only localization and reduces difference between predicted and true posterior distributions in PFN training by more than 50% for the inputs of 5D and 10D[2]. This design leads to substantially lower validation loss and improves predictive performance on high-dimensional regression tasks compared to standard PFN attention, without requiring additional data or compute resources. We prove the localization property of DVA both theoretically and empirically.

- **Attention is More Important than Architecture:** We show that PFNs can also be constructed using CNN as backbone architecture, and with DVA, the choice of backbone architecture becomes secondary. This confirms that the attention mechanism is the primary driver of bias reduction. The proposed CNN-DVA, RNN-DVA and LSTM-DVA based PFN achieves accuracy comparable to a Transformer-DVA based PFN across input dimensions upto 64D. Overall, changes in attention produce a more pronounced reduction in validation loss and predicted error than changes in the backbone architecture. We also show that DVA version of linear attention Choromanski et al. (2022) works considerably better than VA, while being less effective than proposed softmax-DVA.

- **Scaling PFNs to High-Dimensional Learning Problems:** Standard PFNs with joint input–output attention fail to generalize beyond ∼10 input dimensions (10D), saturating at high validation loss. In contrast, DVA enables successful inference up to 64D on power flow learning task. The CNN+DVA achieves Mean Squared Error of order $10^{-5}$ even with 50% load uncertainty levels, and Mean Absolute Error on the order of $10^{-3}$ – at **80×** the speed of exact GP inference.

**Positioning:** We want to highlight that our goal is not to claim a novel, general-purpose attention mechanism. Rather, DVA is a specialized design intended to create scalable and robust PFNs via localization and emulation of GP inference. We also note that there are many efficient attention mechanisms, including linearized kernels Katharopoulos et al. (2020), Nyström approximations Xiong et al. (2021), random feature expansions Choromanski et al. (2021), and cross-kernel attention Wang & Others (2025), which are kernel-based. These attentions are designed to incorporate GP and kernel advantages into Transformer-based language and vision models, along with scaling approximations like Peng et al. (2021); Bui et al. (2025). In contrast, the proposed DVA is designed to develop scalable PFNs Hollmann et al. (2022) that is suitable for physical equations in particular. Further, our sole focus is not on scaling PFNs with Transformer-like Wang et al. (2025), instead a)

---

[1]Here, we use the definition of bias as the difference between the parametrized PPD and the true PPD, with variance vanishing as the number of inference-time samples increases Nagler (2023). Since our experiments operate in this low-variance regime—as also evidenced by the negligible variance observed in our robustness results in Appendix D—any decrease in NLL necessarily reflects a reduction in this bias.

[2]We use D to indicate dimension; for example, ND means N-dimensional.

highlight that attention is the critical component in PFNs over architectures and b) bias reduction in PFNs can be achieved via attention without post-hoc localization Nagler (2023). Since localization of attention for PFNs is the central idea of proposed work, we show that linear attention Choroman-ski et al. (2022) for PFN in decoupled fashion improves performance—relative to coupled linear attention, although softmax-DVA achieves the lowest validation loss. More importantly, DVA is intentionally designed to remain *kernel-free* because forcing a single kernel type can lead to significant model mismatch for physics problems. For instance, the functions governing AC power flow are best modeled by specialized kernels distinct from standard choices (Liu & Srikantha, 2022), and the optimal kernel can even change with operating conditions (Pareek & Nguyen, 2021). By learning a data-driven similarity metric, DVA remains flexible and robust, avoiding the need for manual kernel selection.

## 1.1 RELATED WORKS

**Prior-data Fitted Networks:** There are several works on PFNs (Hollmann et al., 2025; Wang et al., 2025; Nagler, 2023; Adriaensen et al., 2023; Li et al., 2023), most of which rely on the Transformer architecture (Vaswani et al., 2017), applying self-attention over concatenated $(x_i, y_i)$ embeddings. While this design has shown strong performance on certain tasks, it presents two key limitations that remain largely unaddressed. First, these works implicitly assume that the Transformer backbone is crucial to PFN success. Second—and more importantly—the standard attention mechanism does not scale well to high-dimensional problems: training becomes unstable, and performance deteriorates quickly as dimensionality increases Wang et al. (2025). Although Wang et al. (2025) introduced a Boosting-based method that splits the dataset into smaller subsets and trains an ensemble of PFNs, this was primarily intended to handle longer input sequences, not to address high-dimensional scaling issues or architectural dependence of PFNs.

**Physical Equation Surrogates for Power Flow:** Efficiently solving power flow equations is crucial for integrating renewable energy and electric vehicles Barry et al. (2022), a key area where machine learning can help mitigate climate change Rolnick et al. (2022). Faster analytical approximations of nonlinear alternating current power flow (ACPF) equations exist, but come at the cost of accuracy Molzahn et al. (2019). To address this, various ML models—including physics-informed methods—have been developed for ACPF learning and uncertainty quantification Chen et al. (2025). Among these, GPs have gained prominence for building explainable surrogates with closed-form predictions Tan et al. (2025). However, such modeling is extremely sensitive to GP kernels, as shown by Liu & Srikantha (2022) by showing that specialized kernels outperform standard options like squared-exponential or polynomial kernels Pareek & Nguyen (2021).

## 2 BACKGROUND

### 2.1 GAUSSIAN PROCESSES (GP)

GP is a non-parametric, probabilistic framework for modeling functions from a functional space perspective. Given data $(x_i, y_i)$, we assume $y_i = f(x_i) + \varepsilon_i$, where $\varepsilon_i \sim \mathcal{N}(0, \sigma_\varepsilon^2)$. For $N$ inputs $x = (x_1, \ldots, x_N)$, the function values $\boldsymbol{f}(x) = [f(x_1), \ldots, f(x_N)]^\top$ follow a joint Gaussian distribution as $\boldsymbol{f}(x) \sim \mathcal{N}(\boldsymbol{m}(x), K(x, x'))$, with mean function $m(x)$ and covariance matrix $K(x, x')$. By definition, a GP is a collection of random variables such that any finite subset is jointly Gaussian, denoted $f \sim \mathcal{GP}(m(\cdot), k(\cdot, \cdot))$. The observation distribution is then $\mathbb{P}(y) \sim \mathcal{N}(m(x), K(x, x') + \sigma_\varepsilon^2 I)$, with I as identity matrix of appropriate size. Thus, given training data $y$ at $x$, the predictive distribution of $f^\star$ at a new input $x^\star$ is Gaussian with closed-form mean and covariance. The choice of kernel (covariance) function $k(x, x')$ encodes prior assumptions about $f$, while hyperparameters are typically learned by maximizing the marginal log-likelihood. However, the closed-form of exact inference only works when the likelihood is Gaussian and inversion of kernel matrix presents training bottleneck. A key property of GPs, central to the design of DVA, is that the kernel $k(\cdot, \cdot)$ measures similarity solely between input data Williams & Rasmussen (2006).

### 2.2 PRIOR-DATA FITTED NETWORK (PFN)

PFNs (Müller et al., 2022) are neural predictors trained to approximate the *posterior predictive distribution* (PPD) of a Bayesian model in a single forward pass. Rather than fitting a single static

dataset, a PFN is trained on multiple *synthetic datasets*– drawn from a prior over data-generating mechanisms. Given a prior distribution $p(\mathcal{D})$ over supervised learning tasks, PFNs repeatedly sample datasets $\mathcal{D}^k \cup \{(\mathbf{x}^k, \mathbf{y}^k)\} \sim p(\mathcal{D})$ for $k = 1 \ldots K$ and train the model to minimize

$$\text{Negative Log-Likelihood (NLL)} \quad \ell_\theta = \sum_{k=1}^{K} \big[ -\log q_\theta(\mathbf{y}^k \mid \mathbf{x}^k, \mathcal{D}^k) \big]. \tag{1}$$

Here, $q_\theta(\cdot)$ represents the Transformer prediction. This procedure treats entire datasets $\mathcal{D}$'s as inputs and optimizes the model parameters $\theta$ to predict a held-out label conditioned on the remaining data. Thus, fitting the PPD without explicitly computing posteriors. Further, PFNs represent the output distribution using a discrete set of *buckets* (bins) for the target $\mathbf{y}$, essentially posing regression as a classification problem. After training, the PFN performs *amortized Bayesian inference*: given a new dataset $\mathcal{D}_{\text{train}}$ and query point $x_{\text{test}}$, it outputs $q_{\theta^\star}(y_{\text{test}} \mid x_{\text{test}}, \mathcal{D}_{\text{train}}) \approx p(y_{\text{test}} \mid x_{\text{test}}, \mathcal{D}_{\text{train}})$ in a single forward pass, where $\theta^\star$ is optimal Transformer parameters Müller et al. (2022).

## 2.3 LIMITATIONS OF EXISTING PFN ARCHITECTURES WITH JOINT ATTENTIONS

PFNs offer a promising framework for amortized Bayesian inference, though their application to *high-dimensional regression* has so far been relatively limited Hollmann et al. (2025); Wang et al. (2025). The common recipe of using a Transformer backbone that performs self-attention over joint $(\mathbf{x}, \mathbf{y})$ embeddings has a scaling issue. The design choice of representing each training example $(x, y)$ in PFNs as a joint embedding $\text{enc}(x) + \text{enc}(y)$ can be traced back to the way attention-based models have historically treated their basic units of computation. In natural language processing, the Transformer architecture Vaswani et al. (2017) encodes each token as a self-contained representation, i.e. "token as full carrier of information". In this lineage, PFNs adopt the same strategy by concatenating or summing input and output encodings to form a single token while removing positional encoding Müller et al. (2022). Below, we examine the this PFN recipe's structural limitations.

Firstly, attention computation based on the standard PFN initial embedding strategy, of joint representation $\mathbf{z}_i = \text{enc}(\mathbf{x}_i) + \text{enc}(\mathbf{y}_i)$, forces the model to measure across both input features and target output values. As the input dimension grows, pairwise distances concentrate, and the margin between true and spurious neighbors of the input shrinks (the "curse of dimensionality"). Thus, variation in output $\mathbf{y}_i$, unrelated to input proximity, can dominate similarity calculations. Empirically, we observe significant degradation beyond about 10D input in our experiments discussed in Sec. 4.

Further, we can also analyze this dimensionality limitation from a bias perspective as the joint embedding breaks *localization*. The Transformer computes similarity (via dot-product attention) between queries and these mixed embeddings in which the label $\mathbf{y}_i$ contributes equally. This conflicts with theoretical results Nagler (2023), which show that only local samples should influence posterior estimates. Consequently, incorporating joint input–output attention introduces additional bias, which becomes more pronounced in higher dimensions due to the concentration of pairwise distances. In view of these limitations, we propose a simple decoupled value attention (DVA) which keeps the localization intact.

## 3 DECOUPLED-VALUE ATTENTION

We propose **Decoupled-Value Attention (DVA)**, an input-localized attention mechanism for training PFNs. The proposed DVA is structurally aligned with GP inference by treating input $\mathbf{x}$ and output $\mathbf{y}$ separately at the attention stage. We enforce a strict separation of roles: attention affinities (queries and keys) are computed solely from the inputs, while the aggregated information (values) comes from the corresponding outputs– during both PFN training and prediction. Below, we explain DVA mathematically along with comparative assessment against Vanilla Attention (VA) Müller et al. (2022) and a kernel-based attention Wang & Others (2025).

Consider a PFN training dataset $\mathcal{D} = \{X, \mathbf{y}\}$ where $X \in \mathbb{R}^{N \times d}$ and $\mathbf{y} \in \mathbb{R}^{N \times 1}$ with $N$ input samples of dimension $d$. In DVA we calculate query $Q$, key $K$ and value $V$ as

$$Q = W_q \, \varphi_x(X), \quad K = W_k \, \varphi_x(X), \quad V = W_v \, \varphi_y(\mathbf{y}), \tag{2}$$

with encoders $\varphi_x, \varphi_y$ and trainable linear maps $W_q \in \mathbb{R}^{d \times d_k}, W_k \in \mathbb{R}^{d \times d_k}, W_v \in \mathbb{R}^{d \times 1}$. Then, attention is then computed as $\text{Att}(Q, K, V) = \text{softmax}\big(QK^T / \sqrt{d_k}\big) V$. Now, via equation 2, pro-

Table 1: Comparison of Attention Mechanisms for PFNs

| Component | Vanilla | Kernel-based | DVA (ours) |
|---|---|---|---|
| **Input Emb.** | $\text{enc}(x_i) + \text{enc}(y_i)$ | $x_i, y_i$ separately | $x_i, y_i$ separately |
| **Query** | From $z_i$ | $\phi(x_i)$ | From $\text{enc}(x_i)$ |
| **Key** | From $z_j$ | $\phi(x_j)$ | From $\text{enc}(x_j)$ |
| **Value** | From $z_j$ | From $\text{enc}(y_j)$ | From $\text{enc}(y_j)$ |
| **Limitation** | Unstable in high-D | Requires kernel choice | Absent output cues |

Vanilla attention is taken from PFN literature Müller et al. (2022); Hollmann et al. (2022)

posed DVA enforces that similarity is calculated purely in input space, while labels flow only through values. This is unlike VA used in PFNs, which mixes inputs and outputs in a joint embedding.

**Training:** During training we simulate inference by masking one or more labels from the dataset Müller et al. (2022). The unmasked pairs $\mathcal{D}_{\text{cx}} = \{(x_i, y_i)\}_{i=1}^{N_{\text{context}}}$ form the *context set*, while the masked inputs $X_{\text{te}} = \{x_j\}_{j=1}^{M}$ form the *queries*. From the context we build $K_{\text{tr}} = W_k \varphi_x(X_{\text{cx}})$, $V_{\text{tr}} = W_v \varphi_y(\mathbf{y}_{\text{cx}})$. Here, $X_{\text{cx}}$ and $\mathbf{y}_{\text{cx}}$ are matrix and vector forms of $\mathcal{D}_{\text{cx}}$ respectively. Further, from the test (masked) inputs query $Q_{\text{te}}$ in matrix from and labels are predicted by attending $H_{\text{te}}$ to the context as :

$$Q_{\text{te}} = W_q \varphi_x(X_{\text{te}}); \quad H_{\text{te}} = \text{softmax}\left(\frac{Q_{\text{te}} K_{\text{tr}}^\top}{\sqrt{d_k}}\right) V_{\text{tr}}. \tag{3}$$

A head $g(\cdot)$ maps $H_{\text{te}}$ to a predictive distribution, and training minimizes the NLL (equation 1) of the true labels to learn parameters of the network as explained in Müller et al. (2022).

**Inference:** At test time, the mechanism is identical except that *training dataset* forms the *context set* and the "queries" are now the real unseen inputs i.e. we do not know the true output $\mathbf{y}$ for test inputs. Given a training dataset $\mathcal{D}_{\text{train}} \equiv \mathcal{D}_{\text{context}}$ for unseen function learning via GP, we obtain the predicted output with $Q_\star = W_q \varphi_x(X_\star)$ for test input $X_\star$ as

$$\hat{y}_{\text{test}} = g\left(\text{softmax}\left(Q_\star K_{\text{tr}}^T / \sqrt{d_k}\right) V_{\text{tr}}\right) \tag{4}$$

This ensures that the weight assigned to each context point's value $v(y_i)$ depends only on the similarity between the query input $\mathbf{x}_\star \in X_\star$ and the context input $\mathbf{x}_i \in \mathcal{D}_{train}$, mirroring the GP's use of an input-space kernel function, as discussed in the following subsection. The key differences between attention approaches are summarized in Table 1.

### 3.1 LOCALIZATION EFFECT OF DVA AND ALIGNMENT WITH GP INFERENCE

In DVA, the attention weights for a new test point $\mathbf{x}_\star$ are given by $\text{softmax}(\langle Q, K \rangle / \sqrt{d_k})$, where $\langle \cdot, \cdot \rangle$ is the standard dot product. Explicitly, attention weights are

$$\alpha_i(\mathbf{x}_\star) = \frac{\exp\left(\langle W_q \varphi_x(\mathbf{x}_\star), W_k \varphi_x(X_i) \rangle \big/ \sqrt{d_k}\right)}{\sum_{j=1}^n \exp\left(\langle W_q \varphi_x(\mathbf{x}_\star), W_k \varphi_x(X_j) \rangle \big/ \sqrt{d_k}\right)} \tag{5}$$

From equation 5, it is clear that that affinities are determined entirely via relationship between the test input $\mathbf{x}_\star$ and context inputs $X_i$. Unlike joint embeddings $\phi(x, y)$ in VA, the labels $y_i$ do not enter into the similarity measure and only appear downstream through the values as in equation 2. This separation implies that the weight placed on a output $y_i$ depends solely on how well $X_i$ aligns with $\mathbf{x}_\star$ in the projected input space. Thus, the softmax distribution $\alpha_i(\mathbf{x}_\star)$ concentrates mass on a neighborhood of $\mathbf{x}_\star$ because the projection matrices and encoders are trained to align nearby inputs with high value of inner product and push apart distant inputs. Consequently, altering labels (outputs) attached to distant inputs cannot affect the prediction asymptotically, which is exactly the localization property required in Theorem 5.4 of Nagler (2023). This localization property implies that attention weights $\alpha_i(\mathbf{x}_\star)$ associated with context points $\mathbf{x}_i$ that are far from the query $\mathbf{x}_\star$ should contribute negligibly. Below, we first present a theoretical result which proves that linear embedding-based DVA provides attention weights proportional to Mahalanobis RBF kernel using distance between query and context inputs. Then we also show that under linear separability assumptions, nonlinear embedding DVA also exhibit localization behavior. Thus, connecting the proposed DVA mechanism to the formal notion of localization discussed in Nagler (2023).

**Theorem 1 (DVA attention weight $\propto$ Mahalanobis RBF kernel under linear embeddings)**
*Assume the input encoder is linear, i.e. $\varphi_x(\mathbf{x}) = W_x\mathbf{x}$ and the DVA query/key maps are $Q = W_qW_x\mathbf{x}$, $K = W_kW_x\mathbf{x}$. Let $A := (W_qW_x)^\top(W_kW_x)$. If $A$ is symmetric positive definite, and define $\|\mathbf{z}\|_A = \mathbf{z}^T A \mathbf{z}$, then for any query $\mathbf{x}_\star$ and context point $\mathbf{x}_i$ the DVA attention weight equation 5 is proportional to a Mahalanobis RBF kernel*

$$\alpha_i(\mathbf{x}_\star) \;\propto\; \exp\!\left(-\frac{1}{2\tau}\,\|x_\star - x_i\|_A^2\right).$$

**Theorem 2 (DVA localization under nonlinear embeddings)** *Consider a DVA-PFN with fixed query $\mathbf{x}_\star$ and context inputs $\mathbf{x}_i \in \mathcal{D}_{context}$ of size $N_{context}$. Suppose the input encoder $\varphi_x(\cdot)$ is Lipschitz continuous, and that inner product of query $Q_\star$ and key $K_i$ is locally separable around $\mathbf{x}_\star$. Then, as $N_{context} \to \infty$, the total attention mass assigned to all far-away context inputs vanishes with high probability, i.e.: $\sum_{i \in F_\varepsilon} \alpha_i(\mathbf{x}_\star) \xrightarrow[N_{context}\to\infty]{\mathbb{P}} 0$. Where, $F_\varepsilon = \{\, i : \|\mathbf{x}_i - \mathbf{x}_\star\| > \varepsilon \,\}$ denotes the set of "far" indices for any fixed $\varepsilon > 0$.*

Detailed proofs of both these theorems are provided in Appendix B. Beyond the theoretical guarantees, we also provide empirical evidence that, for DVA-based PFNs, the attention weights $\alpha_i$ decay exponentially as the Euclidean distance between the query and context inputs increases. In contrast, VA-based PFNs do not exhibit this behavior (see Fig. 4 and Fig. 5).

We now discuss how DVA aligns with GP inference. As discussed in Section 2.1, GPs model all possible function realizations as zero-mean Gaussians with covariance defined by a kernel, i.e., $f \sim \mathcal{N}(\mathbf{0}, K(X, X'))$, where $X$ is the input and $K(\cdot, \cdot)$ is the kernel matrix over input pairs. Note that parameterization of the possible function family only depends on the input. Further, for a given kernel hyperparameters, the mean prediction $\mu(\cdot)$ of GP is given as a weighted sum of training dataset outputs with weights solely depending upon inputs Williams & Rasmussen (2006):

$$\mu(x_\star) \;=\; \sum_{i=1}^{N_{\text{train}}} \beta_i(x_\star)\, y_i, \quad \text{where } \beta(x_\star) = k(x_\star, X)\left[K(X, X) + \sigma^2 I\right]^{-1}. \tag{6}$$

Following equation 4, 5 and 6, the attention weights in DVA can be interpreted as normalized kernel weights that depend only on the inputs. As in kernel smoothing Tsai et al. (2019), the exponential inner product in $\alpha(\cdot)$ of equation 5 acts as a positive kernel on the input space, with effective bandwidth governed by the scale of the projections and the $1/\sqrt{d_k}$ factor. Thus, similar to GP mean prediction, DVA predictions are obtained as weighted sums of training outputs where the weights are determined entirely by input similarity. Readers can refer to Tsai et al. (2019) for more discussion on the relationship between kernel and attention mechanism.

Here, we want to highlight that DVA's softmax produces non-negative, normalized weights ($\sum \alpha_i = 1$), whereas the GP coefficients $\beta_i(\cdot)$ have no positivity constraint. This limitation is mitigated by subsequent PFN layers (e.g., the final head $g(\cdot)$) and by encoding outputs in the value $V$, which together help adjust the DVA output toward the true GP posterior mean(see Appendix F for effect of final head and value encoder). This construction shows that DVA's architecture implements a predictor of the form "*input-only similarities produce weights, which combine label-dependent values,*" precisely matching the dependency structure of a GP.

Another attention choice for PFNs can be kernel-inspired attentions, which relate GP mean weights $\beta(\cdot)$ and PFN attention weights $\alpha(\cdot)$ more closely– while maintaining input localization by decoupling input and output as in DVA. However, if the input affinities are forced through a fixed kernel function, the PFN will become kernel dependent. As discussed before, identifying the best performing kernel is non-trivial and often requires tailoring kernels to specific function classes Liu & Srikantha (2022). Therefore, it is not advisable to *hard-wire* a particular kernel in PFN design.

To test the effect of kernel dependence on PFN performance, we design a simple Gaussian kernel (radial basis function, RBF) similarity for attention Williams & Rasmussen (2006). We emphasize that this formulation is not equivalent to exact GP kernel regression but rather introduces RBF-style distance-based affinities in place of dot-product similarities Choromanski et al. (2022); Shen et al.

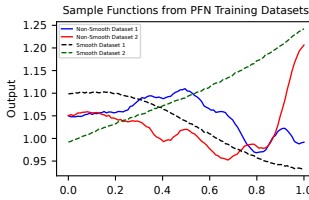 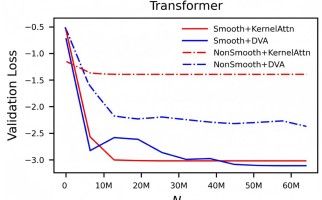 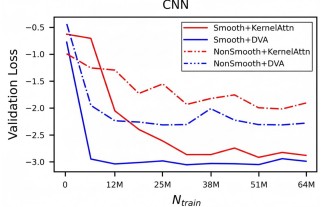

Figure 1: **Effect of Kernel in PFN Attention:** Sample functions from 1D PFN training datasets (Left). Validation loss for smooth and non-smooth functions with Kernel-based Attention and DVA with Transformer (Middle) and CNN (Right).

(2021). The kernel-based attention assigned to a query input $\mathbf{x}_\star$ is then given by

$$\alpha_i(\mathbf{x}_\star) = \frac{\exp\left(-\gamma\left\|W_q\varphi_x(\mathbf{x}_\star) - W_k\varphi_x(X_i)\right\|^2\right)}{\sum_{j=1}^n \exp\left(-\gamma\left\|W_q\varphi_x(\mathbf{x}_\star) - W_k\varphi_x(X_j)\right\|^2\right)} \tag{7}$$

This distance-based attention in equation 7 is more aligned to the RBF kernel; however, it loses flexibility to learn input-localization via training. Thus, the model inherits kernel and $\gamma$ dependence, which may not be suitable for a broader class of functions. To validate this limitation of the kernel-based attention, we test PFN performance with both DVA and this attention. We attempt to learn two classes of functions with different levels of smoothness as shown in Figure 1. The results demonstrate that while kernel-based attention can match DVA in effectively learning smooth functions aligned with the RBF kernel, it under performs on non-smooth functions generated using the linear-periodic kernel. Additionally, we favor DVA over Kernel attention for its superior computational efficiency. By exploiting structural sparsity and hardware-optimized dot products, DVA reduces the computational cost upto four times of dense Kernel attention. We also observed that for RBF training priors, Kernel attention matches performance of DVA while PFN training per epoch is at least four times slower (See Appendix E) for more.

## 4 Numerical Results and Discussion

In this section, we present numerical experiments demonstrating the behavior of PFNs equipped with the proposed DVA and with CNN backbone. The results show that with DVA, PFNs a) train with lower validation loss or residual bias, b) CNN, RNN, LSTM and Transformer perform comparably as architecture, underscoring that attention governs training behavior more than backbone architecture and c) remain scalable for learning in complex physical systems. These findings provide empirical support for the theoretical arguments in Section 3.1. Complete experimental details, including architecture choices, hyperparameter selection, and data generation procedures, are provided in the Appendix C, while additional results are provided in Appendix D.

### 4.1 Bias Reduction and Backbone Architecture Agnostic PFNs

To assess the bias reduction capability of the proposed DVA, we perform PFN learning and testing for datasets of increasing input dimensionality (1D, 2D, 5D, and 10D). Figure 2 plots the validation loss as a function of the training set size $N_{\text{train}}$ for CNN and Transformer backbones equipped with both VA and the proposed DVA.

**Bias Reduction:** Across all input dimensions, the curves with VA (dashed lines) saturate at visibly higher loss values, revealing a persistent residual bias that does not diminish even with large training data. In contrast, DVA-based PFNs (solid lines) consistently converge to lower loss plateaus, demonstrating that DVA mitigates this bias, with negligible increase in variance. The gap becomes especially pronounced in higher dimensions (5D and 10D), where VA-equipped models remain strongly biased, while DVA-equipped models continue to benefit from additional training samples. Further, In the 10D case, we observe an even more striking phenomenon: both CNN+VA and Transformer+VA curves flatten almost immediately after training begins, indicating that the models essentially stop learning. This rapid saturation at high validation loss reflects that VA-equipped PFNs become unable to adapt in higher-dimensional regimes, effectively collapsing to a biased estimator. In contrast, their DVA counterparts continue to decrease loss with additional training data, showing that DVA alleviates this high-dimensional learning obstruction. Another noteworthy trend is visible

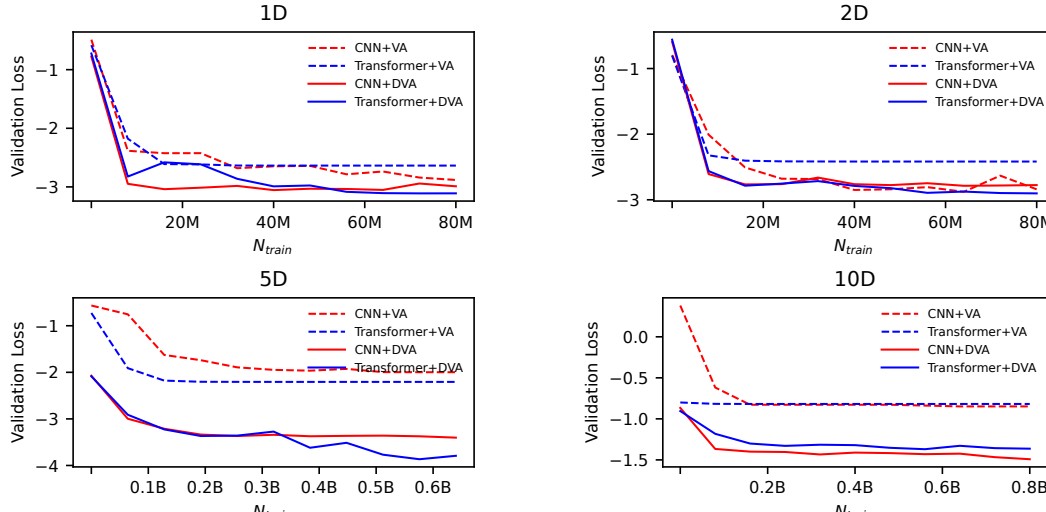

Figure 2: **Bias Reduction in PFN Training:** Validation loss (NLL) behavior with number of training points for various PFNs (Number of training points = epochs × steps per epoch × batch-size × dataset size. Dataset size is 100 for 1D/2D, 400 for 5D and 500 for 10D PFN). Validation loss was calculated on 64 out-of-sample datasets and Transformer + VA is taken from Müller et al. (2022).

at the beginning of training. For low-dimensional tasks (1D and 2D), the initial validation loss is nearly identical across VA and DVA models, with improvements arising only as training progresses. However, in the higher-dimensional cases (5D and 10D), DVA-equipped PFNs already begin with a substantially lower validation loss compared to their VA counterparts, and this advantage compounds as more data are observed. This behavior suggests that DVA not only accelerates convergence but also reduces the asymptotic bias floor, thereby enabling PFNs to faithfully approximate the target physical mappings. To ensure robustness, the 10-dimensional (10D) models were trained multiple times. The corresponding results are provided in the Appendix D.

**Backbone Architecture Agnostic PFNs:** To analyze the effect of backbone architecture on PFN performance, we study 1D, 2D, 5D, and 10D inputs for two network architectures: Transformer Müller et al. (2022) and CNN along with RNN and LSTM backbones on 1D and 10D input spaces. Performance is measured using mean squared error (**MSE**) and validation loss at convergence (**Final Val Loss**), summarized in Table 2. GP results are also included as a baseline for MSE. Each backbone is trained with both VA and the proposed DVA. The results show that attention choice has a larger effect than backbone choice. For instance, at 5D, switching a Transformer from VA to DVA reduces MSE from $2.43 \times 10^{-4}$ to $2.84 \times 10^{-5}$, closer to the GP baseline ($3.42 \times 10^{-6}$). The validation loss also improves from $-2.04$ to $-4.05$—an absolute gain of 2.01 ($\approx 98.5\%$ relative improvement)—while the CNN-Transformer spread under VA is only 0.25 ($\approx 10.9\%$). Similarly, at 10D, CNN and Transformer MSEs drop by nearly an order of magnitude under DVA, far exceeding the architecture gap under VA. These results indicate that CNN- and Transformer-based PFNs perform comparably once the attention mechanism is specified, with DVA further pushing performance toward GP quality in higher dimensions. Additionally, Figure 3 shows that LSTM and RNN backbones can be trained successfully as PFNs, and DVA consistently outperforms vanilla attention in 10D settings across all architectures, while matching the performance in 1D.

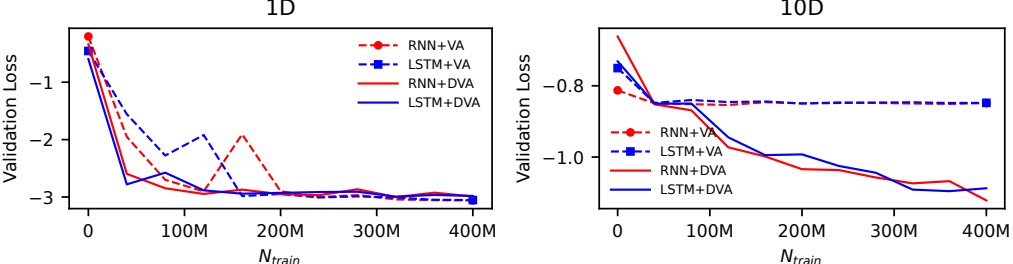

Figure 3: Comparison of validation loss vs. training points ($N_{train}$) for RNN and LSTM architectures with VA and DVA attentions.

Table 2: Mean squared error (MSE) and final validation loss across input dimensions.

| | MSE | | | | | Final Val Loss | | | |
| | GP | VA | | DVA | | VA | | DVA | |
| | | CNN | Tx | CNN | Tx | CNN | Tx | CNN | Tx |
| 1D | 1.02e-04 | 1.07e-04 | 1.28e-04 | 1.37e-04 | 1.23e-04 | $-2.88$ | $-2.63$ | $-3.05$ | $-3.11$ |
| 2D | 1.29e-04 | 1.23e-04 | 1.78e-04 | 2.26e-04 | 1.97e-04 | $-2.91$ | $-2.41$ | $-2.77$ | $-2.91$ |
| 5D | 3.42e-06 | 7.59e-05 | 2.43e-04 | 5.04e-05 | 2.84e-05 | $-2.29$ | $-2.04$ | $-3.56$ | $-4.05$ |
| 10D | 3.47e-04 | 3.55e-03 | 3.56e-03 | 5.49e-04 | 4.98e-04 | $-0.81$ | $-0.81$ | $-1.51$ | $-1.37$ |

Table 3: In-distribution performance of DVA vs. VA across diverse GP priors ($N_{\text{context}} = 80$).

| Training Prior | DVA (MSE) | VA (MSE) | Prior Details ($\ell$: length-scale) |
|---|---|---|---|
| Smooth | **1.90e-04** | 3.16e-04 | Fixed lengthscale: $\ell = 0.25$ |
| Wiggly | **4.19e-03** | 4.26e-02 | Fixed lengthscale: $\ell = 0.03$ |
| Mixed | **1.39e-02** | 1.71e-02 | Sum of two kernels with $\ell_1 \sim U(0.1, 0.5)$; $\ell_2 \sim U(0.01, 0.04)$ |
| All | **2.99e-03** | 2.43e-02 | Per-batch random sampling from the priors above |

**Comparison with GP:** In line with the observations made by authors in Müller et al. (2022), our experiments show that PFNs achieve performance comparable to exact GP inference. As seen in Table 2, PFNs with the proposed DVA consistently move closer to GP performance than those with VA—for instance, in the 10D setting, DVA reduces the MSE from 3.55e-03 (CNN-VA) to 5.49e-04, compared to the GP baseline of 3.47e-04. Importantly, the performance differences between architectures (CNN and Transformer) are relatively minor compared to the gains achieved by changing the attention mechanism, further reinforcing the hypothesis that the effect of attention on PFN performance is far greater than architecture.

We also evaluated the behavior of PFN inference as a function of $N_{\text{context}}$, i.e., *How PFN performance improves as the number of available samples increases at inference time?*. As shown in Figure 7, PFNs (both CNN and Transformer-based) with the proposed DVA exhibit a consistent decrease in error with increasing context, closely matching the performance of exact GPs in low-dimensional settings. In higher dimensions, GPs maintain a slight advantage, consistent with the trends observed in the training performance analysis. It is important to note that the observed performance gap between 5D and 10D (for both PFNs and GP) arises largely because of limited samples per dataset for 10D model (400 for 5D and 500 for 10D). Similar plots for 1D, 2D PFNs MSE, along with MAE and maximum error for all-dimensional PFNs, are given in the Appendix D.

To test DVA's robustness beyond a single function class, we trained and evaluated both DVA and VA on three distinct GP priors generating Smooth, Wiggly (high-frequency), and Mixed complexity functions from RBF kernel. Table 3 shows that DVA consistently outperforms VA across all priors. This confirms DVA's architecture is robust and not over-tuned to a specific prior. The higher MSEs for the Wiggly and Mixed priors are expected, as their functional complexity presents an intrinsically more challenging learning task for a fixed context size and number of training points.

**Uncertainty Quantification & Post-Hoc Localization**: In the Appendix H we preovide results indicating that proposed DVA PFN can provide calibrated predictive uncertainty distributions while in Appendix K we compare post-hoc localization Nagler (2023) with the proposed method. Results indicate that in low-dimensions post-hoc localization reduces error and can even slightly outperform DVA. However, in higher-dimensions where VA PFN doesn't get trained, localization provides no benefit. Thus, emphasizing the need of DVA's architectural locality.

## 4.2 PHYSICS EQUATION LEARNING

**Rosenbrock Function:** To benchmark our method against a well–established baseline, we conduct experiments on the 5-dimensional Rosenbrock function Rosenbrock (1960), a standard test problem in optimization that is often interpreted as a nonlinear potential energy landscape with a curved valley structure Akian et al. (2022). GPs are a natural choice for such comparisons because they provide a flexible non-parametric model with uncertainty quantification, and they have been widely

Table 4: **Voltage prediction on a 64D power-flow test bed:** Trained on 500 samples; evaluated on 4,500 test samples. The time results ($t$) are for evaluating on all 32 node voltages, and the MSE and MAE correspond to the maximum values across the buses.

| | Exact GP | | | CNN + DVA | | | Transformer + DVA | | |
|---|---|---|---|---|---|---|---|---|---|
| $\Delta$**Load** | **MSE** | **MAE** | $t$ | **MSE** | **MAE** | $t$ | **MSE** | **MAE** | $t$ |
| 5% | 2.2e-7 | 0.0004 | 10.88 | 4.5e-7 | 0.0005 | 0.13 | 1.5e-6 | 0.001 | 0.17 |
| 10% | 3.5e-7 | 0.0004 | 10.94 | 1.7e-6 | 0.001 | 0.13 | 2.8e-6 | 0.001 | 0.17 |
| 30% | 3.2e-7 | 0.0005 | 11.61 | 1.5e-5 | 0.003 | 0.14 | 1.6e-5 | 0.003 | 0.17 |
| 50% | 2.2e-7 | 0.0003 | 11.89 | 4.2e-5 | 0.005 | 0.13 | 4.4e-5 | 0.005 | 0.17 |

benchmarked on Rosenbrock and related test functions in the GP literature Xu et al. (2025). Results indicate that 5D PFN with Transformer+DVA shows MSE 6.8e-4 and CNN+DVA achieves MSE of 1.6e-3, without any retraining, see Table 9 in Appendix D for detailed results.

**Power Flow Learning:** In this experiment, we model the IEEE 33-bus distribution system by treating the real and reactive power demands at each of the 32 load buses as uncertain inputs (same experiment design as described in Pareek & Nguyen (2021); Liu & Srikantha (2022)). This results in a 64-dimensional input space (32 active + 32 reactive loads). Now the learning task is to predict the corresponding steady-state bus voltage magnitude— effectively learning the nonlinear AC power flow mapping from loads to voltages i.e. Voltage $= f$(Loads) (See equation 9 in Appendix A.2). Table 4 benchmarks power flow surrogates under varying load perturbations from 5% to 50%. Exact GP achieves the lowest MSE and MAE values across all cases, but requires training 32 times (one for each node), which becomes *infeasible for repeated queries under changing load conditions*. In contrast, both PFNs CNN+DVA and Transformer+DVA trade a modest increase in error for dramatic efficiency gains—over $80\times$ faster than GPs—while maintaining voltage prediction accuracy at the order of $10^{-3}$, sufficient for practical grid analysis. Further, the prediction error decreases as more training (context) samples are provided, with both CNN+DVA and Transformer+DVA converging to near-identical performance as illustrated in Figure 9 (Appendix D). These results highlight that while GPs remain the gold standard for accuracy, DVA-equipped PFNs offer a scalable alternative, enabling high-dimensional, uncertainty-aware power flow learning in real time for complex networked systems. Moreover, because voltages are in per-unit, MSE and MAE values around $10^{-3}$ are practically acceptable. In real systems, measurement devices typically have least counts of $10^{-3}$ p.u., so an error of $10^{-3}$ in a 1 kV system corresponds to just 1 V Molzahn et al. (2019). We also note that, consistent with the 10D case in Figure 2, PFNs equipped with vanilla attention failed to train sufficiently for this 64D problem and thus did not yield meaningful results. Training time for 64D models is approximately 14 hours for both Transformer and CNN-based PFNs on NVIDIA 4500ADA GPU.

## 5 CONCLUSIONS AND FUTURE WORK

In this work, we propose Decoupled-Value Attention (DVA) to train Prior-Data Fitted Networks (PFNs), particularly for GP inference for high-dimensional inputs. Through experimental studies, we show that the proposed DVA halves the residual bias in PFN learning for 5D and 10D settings, and enables PFNs constructed with either CNNs or Transformers to achieve comparable accuracy once equipped with the attention mechanism. Leveraging these advantages, DVA enables PFNs to serve as highly efficient surrogates for high-dimensional power flow learning. On the IEEE 33-bus system with 64-dimensional load variations, DVA-equipped PFNs attained voltage prediction accuracy in the order of $10^{-5}$ while delivering more than an $80\times$ speedup over exact GP. Our analysis, grounded in Nagler (2023) locality requirement, shows that input-only attention offers a principled advantage over joint $(x, y)$-attention in settings where consistent inference demands vanishing influence from far-away inputs. By removing label-driven cross-terms, DVA enforces this locality, while its strict input focus may be limiting in inherently nonlocal tasks (e.g., long-range time-series like weather). Thus DVA clarifies when input-local attention is beneficial and when joint attention may still be necessary. Future work will focus on scaling PFNs to even larger power networks and higher-dimensional uncertainty spaces, particularly in the context of power flow uncertainty quantification and planning problems. Together, these efforts can push PFNs toward practical deployment for real-time, uncertainty-aware decision making in modern power systems.

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

# Appendix

## A   PHYSICS EQUATION BENCHMARKS

### A.1   ROSENBROCK FUNCTION

For our baseline experiments, we use the 5D–dimensional Rosenbrock function, defined as Rosenbrock (1960)

$$f(\mathbf{x}) = \sum_{i=1}^{d-1} \left[ 100 \left( x_{i+1} - x_i^2 \right)^2 + (1 - x_i)^2 \right], \quad \mathbf{x} \in [-1, 1]^d, \ d = 5. \tag{8}$$

We normalize the input vectors $\mathbf{x}$ to the unit hypercube $[0, 1]^5$ before training, and outputs are standardized using $Z$-score normalization. For GP testing, we employ a Gaussian process surrogate with a RBF kernel with automatic relevance determination (ARD) length-scales Williams & Rasmussen (2006).

### A.2   AC POWER FLOW PROBLEM

The alternating current power flow (ACPF) problem is fundamental to power grid analysis, as it computes the steady-state voltages, currents, and power flows that satisfy Kirchhoff's laws under given nodal injections. Unlike ACOPF, which optimizes generator set-points, ACPF focuses on feasibility by solving the nonlinear power flow equations, which are given as:

$$P_i = \Re \left\{ V_i \sum_{j \in N} Y_{ij}^* V_j^* \right\}, \quad Q_i = \Im \left\{ V_i \sum_{j \in N} Y_{ij}^* V_j^* \right\}, \quad \forall i \in N, \tag{9}$$

where $P_i$ and $Q_i$ are the real and reactive power injections at bus $i$, $V_i$ is the complex bus voltage, and $Y_{ij}$ are the elements of the bus admittance matrix.

In our setting, we explicitly consider uncertainty at each bus in both real and reactive power injections. For an IEEE 33-bus system Pareek & Nguyen (2021); Liu & Srikantha (2022), with the first bus designated as the slack bus (zero load), this leads to a 64-dimensional uncertainty input vector capturing nodal variability across all other buses. We follow the standard ACPF model used in `PowerModels` Coffrin et al. (2018), and we use `compute_ac_pf` function of `PowerModels.jl` to generate dataset.

#### A.2.1   POWER FLOW LEARNING WITH GPS

In the power flow learning setting, the goal is to approximate the mapping from net load vectors to system states such as bus voltages (magnitude and angle). This mapping, though implicitly defined by the nonlinear power flow equations (equation 9), is treated here as a supervised regression task where the net load serves as input and the voltage response as output.

We adopt a GP model to capture this relationship:

$$y(\boldsymbol{x}) = f_s(\boldsymbol{x}) + \varepsilon, \quad \varepsilon \sim \mathcal{N}(0, \sigma_\varepsilon^2), \tag{10}$$

where $y(\boldsymbol{x})$ is the observed voltage at a node for load vector $\boldsymbol{x}$. With GP priors, $\boldsymbol{y}(\boldsymbol{x}) \sim \mathcal{GP}(0, K(\boldsymbol{x}, \boldsymbol{x}) + \sigma_\varepsilon^2 I)$, and the kernel $K$ encodes correlations between operating points. Owing to the smoothness of voltages as a function of load, the squared exponential kernel has been widely used in prior work Tan et al. (2025).

GP-based approximations have been shown to outperform analytically approximated linearized models in capturing power flow uncertainty Pareek & Nguyen (2021) and are favored over other learning methods such as DNN due to closed-form approximation nature of GP, and predictive variance availability etc. For more details on power flow modeling and GP surrogate of it, readers can refer to Tan et al. (2025).

# B    PROOF OF DVA LOCALIZATION THEOREM

This section provides the full proof of Theorem 1 and Theorem 2 and an additional corollary covering the linear-embedding case, where the DVA logit are shown to be a Mahalanobis RBF kernel.

**Proof of Theorem 1.**    Assume $u(\mathbf{x}) = W_q W_x \mathbf{x}$ and $v(\mathbf{x}) = W_k W_x \mathbf{x}$, and define $A = (W_q W_x)^\top (W_k W_x)$. Consider $A$ is symmetric and positive definite (e.g. Consider a case where $W_q = W_k$ with full rank $W$'s), then for any context point $\mathbf{x}_i$,

$$\ell_i = \langle Q_\star, K_i \rangle = \mathbf{x}_\star^\top A \mathbf{x}_i = \frac{1}{2}\big(\|\mathbf{x}_\star\|_A^2 + \|\mathbf{x}_i\|_A^2 - \|\mathbf{x}_\star - \mathbf{x}_i\|_A^2\big),$$

with $\|z\|_A^2 = z^\top A z$ being energy norm. Exponentiating and dividing by temperature $\tau$ ($\sqrt{d_k}$ in our case) gives

$$\exp(\ell_i/\tau) = \exp\Big(\frac{\|x_\star\|_A^2 + \|x_i\|_A^2}{2\tau}\Big) \cdot \exp\Big(-\frac{\|x_\star - x_i\|_A^2}{2\tau}\Big).$$

Up to per-point norm factors, the attention weight from equation 5 satisfies

$$\alpha_i(\mathbf{x}_\star) \propto \exp\big(-\tfrac{1}{2}\|\mathbf{x}_\star - \mathbf{x}_i\|_A^2\big).$$

Hence, DVA reduces to a Mahalanobis RBF kernel and is therefore automatically local: the attention weights decay exponentially with distance.    □

**Proof of Theorem 2.**    Fix a query $\mathbf{x}_\star$ and context inputs $\{\mathbf{x}_i\}_{i=1}^{N_{\text{context}}}$ Let, $u(x) = W_q\,\varphi_x(x)$, $v(x) = W_k\,\varphi_x(x)$. Consider the query-key inner product as:

$$\ell_i = \langle u(\mathbf{x}_\star), v(\mathbf{x}_i)\rangle, \qquad \alpha_i(\mathbf{x}_\star) = \frac{\exp(\ell_i)}{\sum_{j=1}^{N_{\text{context}}} \exp(\ell_j)}.$$

For a fixed $\varepsilon > 0$, define the sets of near and far indices

$$N_\varepsilon = \{i : \|\mathbf{x}_i - \mathbf{x}_\star\| \le \varepsilon\}, \qquad F_\varepsilon = \{i : \|\mathbf{x}_i - \mathbf{x}_\star\| > \varepsilon\}.$$

By the local-separation of inner product assumption, there exist constants $\gamma \in \mathbb{R}$ and $\delta > 0$ such that $\ell_i \ge \gamma\,(i \in N_\varepsilon)$, and $\ell_j \le \gamma - \delta\,(j \in F_\varepsilon)$. Hence, considering each exponent at bound

$$\sum_{j \in F_\varepsilon} \exp(\ell_j) \le |F_\varepsilon|\,e^{\gamma - \delta}, \qquad \sum_{i \in N_\varepsilon} \exp(\ell_i) \ge |N_\varepsilon|\,e^{\gamma}.$$

Using these inequalities in the definition of the softmax weights yields

$$\sum_{j \in F_\varepsilon} \alpha_j(\mathbf{x}_\star) = \frac{\exp(\ell_i)}{\sum_{i \in N_\varepsilon} \exp(\ell_i) + \sum_{j \in F_\varepsilon} \exp(\ell_j)} \le \frac{\exp(\ell_i)}{\sum_{i \in N_\varepsilon} \exp(\ell_i)} = \frac{|F_\varepsilon|\,e^{\gamma - \delta}}{|N_\varepsilon|\,e^{\gamma}} = \frac{|F_\varepsilon|}{|N_\varepsilon|}\,e^{-\delta}.$$

Therefore, as context length goes to infinity $N_{\text{context}} \to \infty$, local samples $|N_\varepsilon| \to \infty$ in probability while $|F_\varepsilon|/|N_\varepsilon|$ remains bounded ($F_\varepsilon$ is complement of $N_\varepsilon$). Therefore the right-hand side converges to 0 in probability, and thus

$$\sum_{j \in F_\varepsilon} \alpha_j(\mathbf{x}_\star) \xrightarrow[N_{\text{context}} \to \infty]{\mathbb{P}} 0.$$

This proves that DVA attention becomes fully input-local as context size grows.    □

**Note[Local Separability Assumption]:** The local-separability assumption is generally mild and naturally satisfied when the encoder $\varphi_x$ preserves the local geometry of the input space, i.e., when nearby inputs remain close in the embedding and distant inputs remain well separated. This typically holds for smooth, Lipschitz neural encoders such as MLPs or Fourier-feature mappings applied to regression tasks with locally regular structure. In contrast, the assumption may fail when the underlying task or encoder is inherently *nonlocal*—for example, when the target function is periodic, symmetric, or globally dependent, or when the encoder distorts geometry through aliasing or collapse. Simple cases include encoder *collapse* (e.g., when nonlinearities saturate and map many inputs to nearly the same vector) and *extreme compression* (e.g., a narrow bottleneck that forces very different inputs to share the same low-dimensional code). In these situations, distant inputs may appear artificially similar to the query, breaking the monotonic relationship between distance and attention.

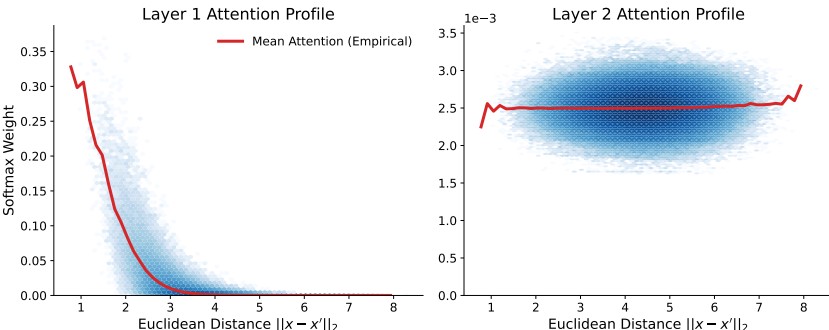

Figure 4: In the first layer, we can clearly see that as the Euclidean distance increases, the softmax weight decreases exponentially, clearly showing that the DVA mechanism enforces localization. Localization is enforced in the first layer, and since layer 2 is the last layer of the model, which outputs the exact values, the softmax values are all minimal and closer to each other, suggesting the last layer averages the result for proper PPD approximation. Essentially, Layer 1 performs the "Local Smoothing" (gathering information from neighbors). Layer 2 performs "Feature Mixing" (processing the gathered information). Since Layer 1 has already gathered the local information into the latent vector, Layer 2 no longer needs to be spatially local; it can attend globally or uniformly to refine the prediction. This provided empirical evidence of the DVA localization theorem.

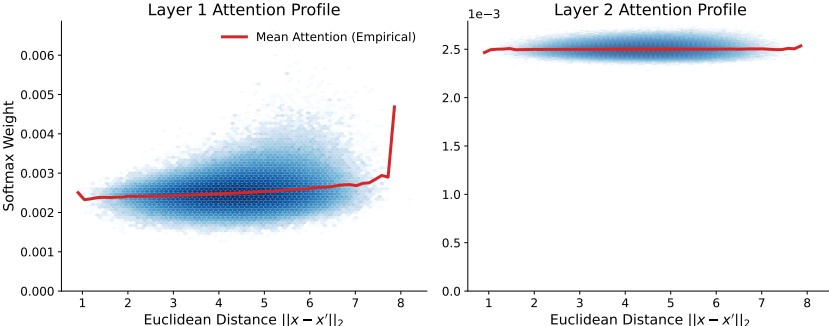

Figure 5: In contrast to the DVA, the standard VA Transformer assigns near-uniform attention weights across the entire input domain (flat trend lines). This confirms that without explicit localization through architecture changes (which we made in DVA), the model defaults to global averaging rather than local interpolation. Attention weights in both layers remain effectively uniform regardless of input distance, empirically validating Theorem 6.3 of Naglar 2023.

## C  IMPLEMENTATION AND ARCHITECTURAL DETAILS

**Synthetic Prior Data Generation:** To assess different attention mechanisms in Prior-Data Fitted Networks (PFNs), we use synthetic datasets generated from GP priors, following Müller et al. (2022). Each regression dataset consists of input–output pairs $(x, y)$, with inputs sampled uniformly and outputs drawn from a multivariate Gaussian with an RBF kernel having lengthscale, variance (output scale), and observation noise variance as hyperparameters. The inputs are normalized using Z-score normalization, while outputs are shifted to a range of $0.8$–$1.2$ for all datasets.

For classification-based objectives, the continuous output space is discretized into buckets derived from quantiles of GP-sampled outputs Müller et al. (2022). Each bucket corresponds to a categorical class index, allowing regression-style PFN training to be cast into classification under a Riemannian distribution loss formulation. This strategy preserves ordering structure while making outputs compatible with categorical training setups. See Müller et al. (2022) for more information in this.

Table 5: Number of buckets for different input dimensions PDFs.

| Dimensions | 1D | 2D | 5D | 10D | 64D |
|---|---|---|---|---|---|
| Number of Buckets | 100 | 100 | 500 | 500 | 500 |

**Transformer Architecture:** We used a Transformer architecture where input features and targets are first projected into a shared embedding space and then processed through a series of encoder blocks combining attention, feedforward layers, residual connections, and layer normalization. The model's hyperparameters, includes the model width (embedding size), number of attention heads, number of encoder blocks, and hidden dimension of the feedforward layers and all parameters are initialized using Xavier uniform initialization.

**CNN-Attention Architecture:** The CNN-attention model also encodes features and targets into a shared embedding space using linear layers. The embeddings are then processed through a stack of convolutional-attention blocks, where each block applies single dimension depthwise convolutions followed by attention, combined with residual connections and layer normalization. Finally, a small DNN head maps the processed embeddings to the model outputs. Key hyperparameters are model width, number of layers, and kernel size.

**Hyperparameters Selection:** To ensure fair evaluation across architectures and embedding dimensions, we employed `Optuna` Akiba et al. (2019) for automated hyperparameter tuning. Key parameters such as model width, hidden dimension size, number of attention blocks, number of heads, and dropout rate were jointly optimized, with `AdamW` and a linear warmup followed by step-wise decay. Each trial involved computing training and validation losses on PFN tasks, and `Optuna`'s pruning strategies enabled efficient exploration of the search space. The best-performing configurations were selected based on initial validation loss over 1000 trials, while training loss was also tracked to assess stability.

Table 6: Transformer Hyperparameter search ranges used in `Optuna`.

| Model Width | Hidden Dim | Attention Blocks | Heads | Dropout |
|---|---|---|---|---|
| 32–256 | 128–1024 | 1–4 | 2–8 | 0.0–0.5 |

Table 7: CNN Hyperparameter search ranges used in `Optuna`.

| Model Width | Layers | Kernel Size |
|---|---|---|
| 32–256 | 1–6 | 3, 5, 7 |

Table 8: Number of trainable parameters across input dimensions (same for VA and DVA).

| Model | 1D | 2D | 5D | 10D |
|---|---|---|---|---|
| CNN | 9,060 | 9,092 | 36,116 | 36,276 |
| Tx | 316,645 | 316,673 | 878,198 | 688,854 |

Table 9: Comparison of MSE values for different models with increasing training points for Rosenbrock Function approximation.

| Training Points | GP | CNN+DVA | Transformer+DVA |
|---|---|---|---|
| 10 | 1.02e-2 | 8.65e-3 | 9.12e-3 |
| 50 | 5.76e-3 | 5.41e-3 | 4.01e-3 |
| 100 | 3.92e-4 | 4.13e-3 | 2.41e-3 |
| 200 | 7.70e-5 | 3.06e-3 | 1.84e-3 |
| 500 | 1.00e-7 | 1.61e-3 | 6.83e-4 |

# D    ADDITIONAL RESULTS

Figure 6 presents the validation loss curves for five different models trained on the 10-dimensional input setting, plotted against the number of training samples $N_{train}$. Each curve represents the mean validation loss over six independent training runs, with shaded regions indicating the minimum and maximum loss values across these runs, illustrating the variability and robustness of the training process. As observed, the CNN + VA and Transformer + VA models show poorer training performance, consistent with the results discussed in the main paper. In contrast, the CNN + DVA and Transformer + DVA models exhibit significantly improved and more stable training behavior. These findings highlight the robustness of our implementation.

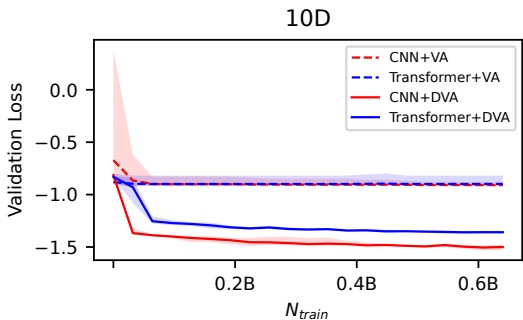

Figure 6: Robustification study for training 10D PFNs. Curves show the mean validation loss over 6 runs; shaded regions represent the minimum and maximum loss values across runs.

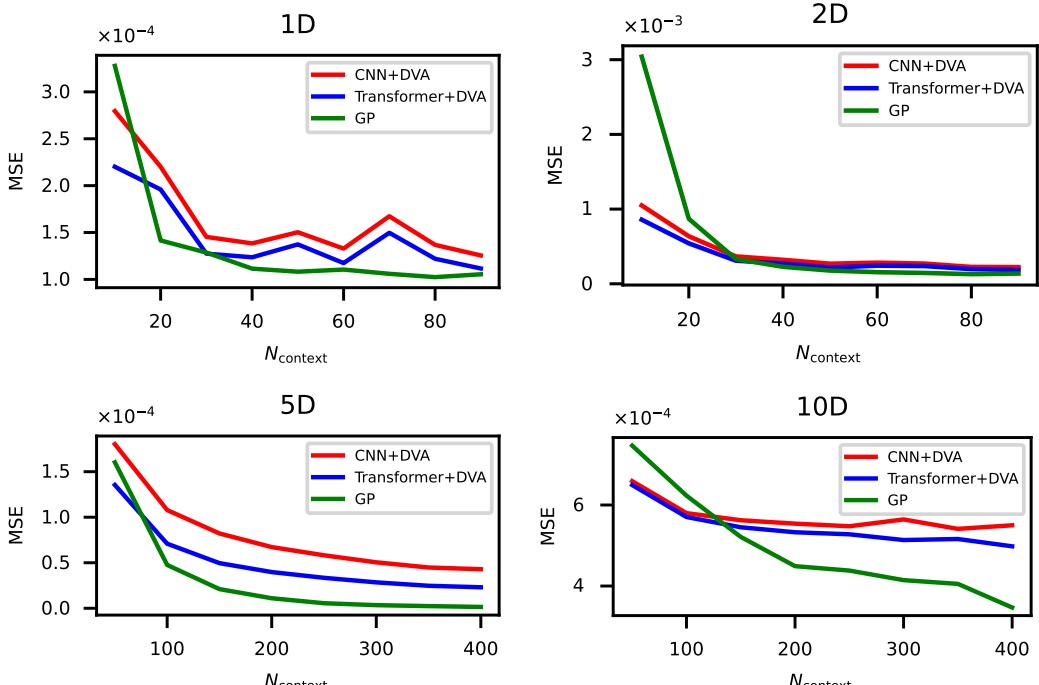

Figure 7: **Comparison with GP:** MSE for 1D,2D, 5D and 10D PFNs as a function of context size. All models are tested using $n_{\text{test}} = 500$, for $N_{context}$. The results show that error consistently decreases with larger context sizes, and that CNN- and Transformer-based PFNs with DVA approach the performance of exact GP inference even in higher dimensions. Exact GP baselines were fit using `scikit-learn` with $N_{context}$ training samples.

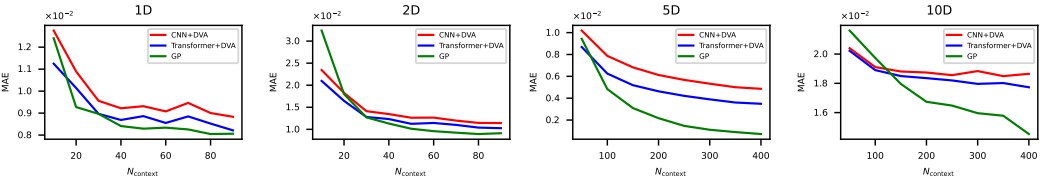

Figure 8: **MAE for PFNs across context sizes:** Mean absolute error as a function of $N_{\text{context}}$. Models were tested with $n_{\text{test}} = 500$ points per dataset. 1D and 2D PFNs were trained with 100, while 5D and 10D PFNs used 500 points per dataset. Error decreases with larger context sizes, and CNN- and Transformer-based PFNs with decoupled-value attention (DVA) approach the performance of exact GP, even in higher dimensions. Exact GP baselines were fit using `scikit-learn`.

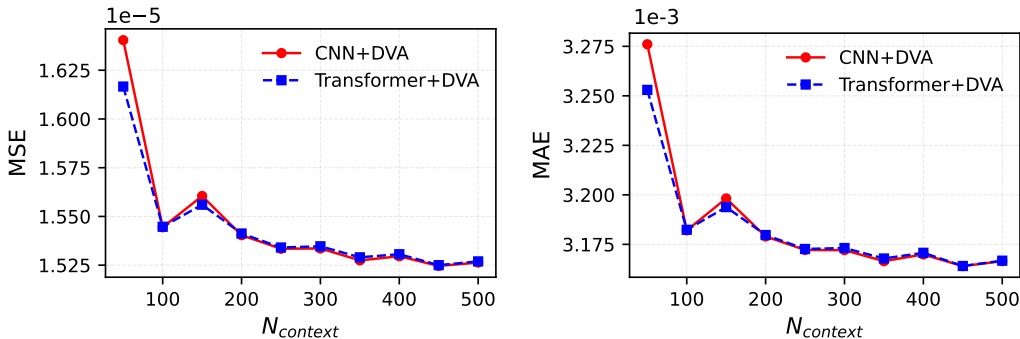

Figure 9: **Learning performance on the 64D power-flow task:** The plots show variation of MSE (**left**) and MAE (**right**) with the number of training context samples ($N_{\text{context}}$). Both CNN+DVA and Transformer+DVA exhibit decreasing errors with additional context and converge to near-identical accuracy. Testing is performed on 4500 out-of-sample testing data of voltages.

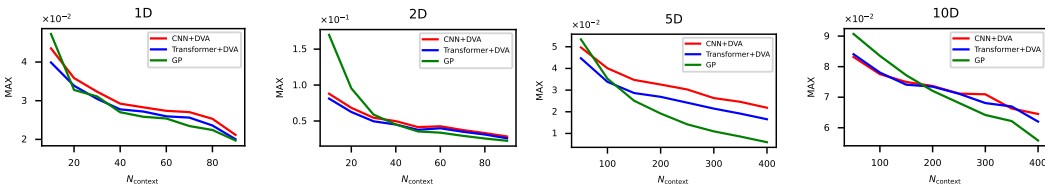

Figure 10: **Maximum error for PFNs across context sizes:** Maximum error as a function of $N_{\text{context}}$. Models were tested with $n_{\text{test}} = 500$ points per dataset. 1D and 2D PFNs were trained with $n = 100$, while 5D and 10D PFNs used $n = 500$ points per dataset. Error decreases with larger context sizes, and CNN- and Transformer-based PFNs with decoupled-value attention (DVA) approach the performance of exact GP, even in higher dimensions. Exact GP baselines were fit using `scikit-learn`.

## E    KERNEL-BASED ATTENTION EXPERIMENT DETAILS

For the experiments shown in Figure 1, we generated synthetic datasets using both smooth and non-smooth kernels, and trained 1D Transformer and CNN models (with the same backbone architectures as used in Table 2) for both kernel attention and DVA. The smooth datasets were sampled from an RBF kernel, which promotes locality and smoothness in the function. In contrast, the non-smooth datasets were generated using a linear–periodic kernel as discussed in Duvenaud (n.d.), which combines a linear trend with a periodic component, producing oscillatory patterns with irregular variations and reduced smoothness. Our results confirm that while Kernel-based attention and the proposed DVA reach effectively the same performance level on RBF training priors, Kernel attention requires significantly more computational time to do so. Quantitatively, both methods achieve comparable predictive error on 1D smooth functions (MSE of $1.40 \times 10^{-4}$ for Kernel vs. $1.90 \times 10^{-4}$ for DVA) and converge to similar validation losses on the 10D task ($-1.50$ for Kernel vs. $-1.29$ for DVA). However, the training logs highlight a drastic difference in efficiency: DVA is approximately $4\times$ faster, completing 80,000 optimization steps in just 42 minutes, whereas the Kernel model managed only 23,000 steps in a longer duration of 48 minutes, demonstrating that DVA delivers the performance of kernel methods with vastly superior throughput.

## F    ABLATION STUDIES

### F.1    ABLATION OF THE VALUE ENCODER $\phi_y$

This experiment studies whether a learnable value encoder $\phi_y$ is required and whether higher-capacity encoders improve GP-style inference in DVA-PFNs. A simple linear encoder from $y$ to the

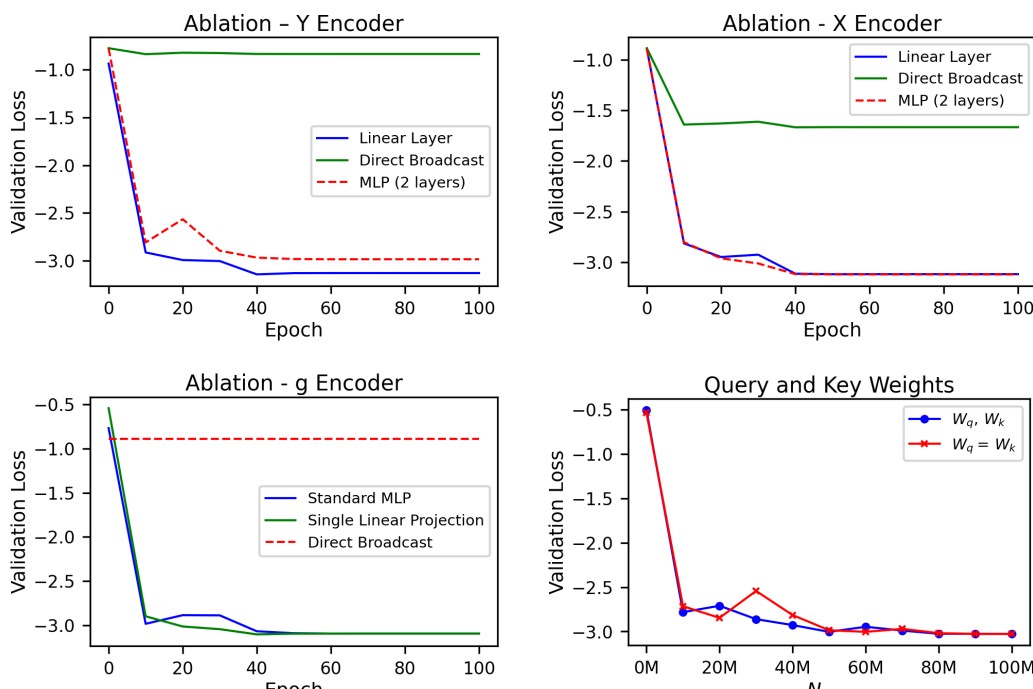

Figure 11: **Ablation results for encoders $\phi_y$, $\phi_x$, the prediction head $g$, and query/key projections $W_q, W_k$.** Linear encoders for both $x$ and $y$ perform as well as deeper MLP-based encoders, whereas direct broadcasting produces large degradation. Learnable $\phi_y$ and $g$ are essential for mitigating the mismatch between GP regression weights (which may be positive or negative) and softmax attention weights (strictly positive). Finally, tying $W_q = W_k$ performs nearly identically to learning them independently, indicating that DVA-PFNs are robust to the choice of kernel-projection parameterization. Here $N_{\text{train}}$ is measured in millions.

model dimension performs the best, matching a 2-layer MLP while substantially outperforming a direct broadcast baseline. These results highlight that a learnable value encoder is essential for mitigating the mismatch between GP regression weights (which can be positive or negative) and softmax attention weights (strictly positive). The encoder $\phi_y$ provides the necessary representational flexibility so that the model can emulate signed GP-like contributions through feature transformations rather than through the attention weights alone. Thus, although simple, the linear $\phi_y$ is necessary for effective DVA aggregation.

## F.2 ABLATION OF THE INPUT ENCODER $\phi_x$

We evaluate whether a learnable input encoder is essential for representing the geometry implied by the GP prior. Both the linear encoder and the 2-layer MLP encoder perform nearly identically (validation loss $\approx -3.12$), indicating that moderate capacity is enough for expressing the input geometry required by DVA. Removing the encoder and broadcasting $x$ across channels significantly degrades performance (validation loss $\approx -1.68$), demonstrating that DVA needs a trainable representation but not additional encoder depth.

## F.3 ABLATION OF THE PREDICTION HEAD $g(\cdot)$

We ablate the prediction head $g(\cdot)$, which converts the final hidden representation into bucketized predictive logits. Both the MLP head and the simple linear head converge near $-3.1$, indicating that additional nonlinearity does not improve predictive performance. However, removing all learnable parameters and broadcasting a single hidden dimension to all output bins fails completely (validation loss $\approx -0.90$). A learnable $g$ is therefore required for the same reason as $\phi_y$: GP regression weights

can be positive or negative, while softmax attention weights cannot. The output head compensates by reparameterizing hidden features so that the model can emulate signed GP-like effects through linear combinations of positive-weighted attention outputs. Thus, any learnable $g(\cdot)$ is sufficient, but some learnability is essential.

### F.4 ABLATION OF QUERY/KEY PROJECTIONS $W_q, W_k$

This ablation tests whether DVA depends on having independent query and key projections. We compare the standard configuration with separate $W_q$ and $W_k$ to a shared-projection variant where $Q = K = W_{qk}(x)$. Across all training-set sizes (with $N_{train}$ measured in millions), both variants perform nearly identically, converging near $-3.0$. These findings show that **DVA-PFN performance is not sensitive to whether query and key projections are shared or independent**. The model appears to operate primarily through the relative input geometry encoded in $\phi_x$, not through asymmetry introduced by $W_q$ and $W_k$. This suggests redundancy in the standard attention parameterization and further indicates robustness of DVA to architectural choices.

### F.5 BUCKET SIZE ABLATION

For both CNN+DVA and Transformer+DVA, increasing the number of output bins improves validation loss in both 1D (left column) and 2D (right column) tasks. The improvement is sharp when moving from small to moderate bin sizes, and performance stabilizes once the bin size reaches roughly 150–200 bins. Overall, DVA-PFNs are not highly sensitive to bin size, as long as it is sufficiently large to provide adequate resolution of the predictive distribution.

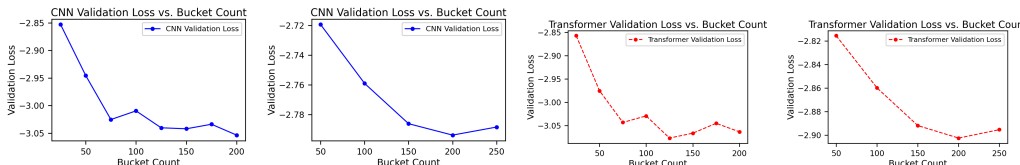

Figure 12: **Effect of Bin Size on Validation Loss:** Validation loss variation with different bin sizes for CNN+DVA (top row) and Transformer+DVA (bottom row) across 1D (first column), and 2D (second column).

## G LINEAR ATTENTION

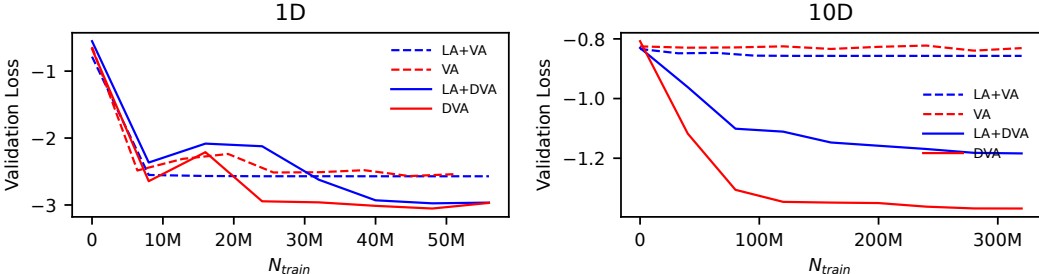

Figure 13: Comparison with performer/linear attention using Transformer PFN. Linear attention is used with both VA and DVA. As is visible, the decoupling is allowing for better training; the effect is visible clearly in 10D. Softmax DVA performs best as expected, followed by Linear approximation of DVA. VA performs worse, and is unable to train in 10D.

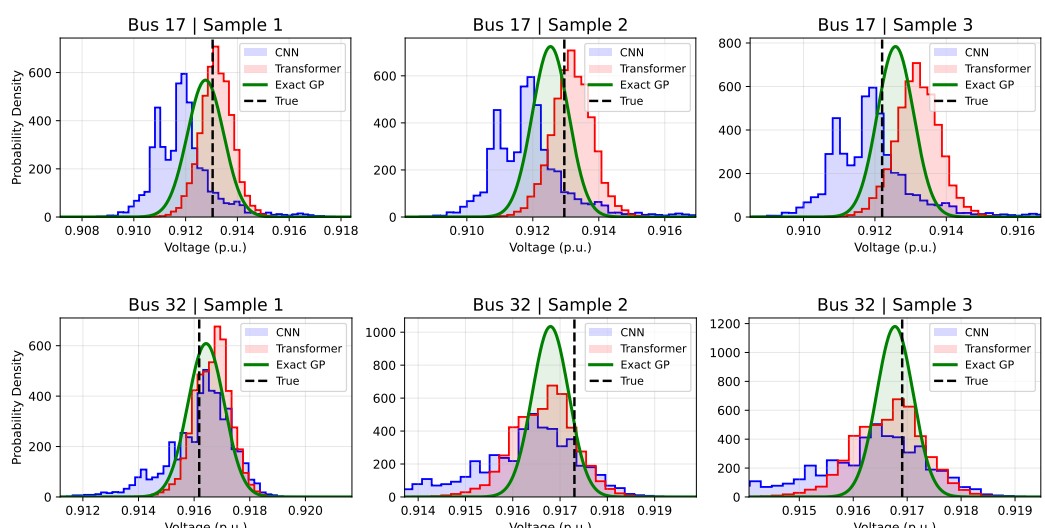

Figure 14: Comparing the PPD for three distinct samples for (Top) Bus 18. (Bottom) Bus 33

Table 10: Percentage of values within variance bounds node 32 (farthest node from Generator bus) and 4500 testing samples (500 Context/Training points)

| Model | $\pm 0.1\sigma$ | $\pm 1\sigma$ | $\pm 2\sigma$ |
|---|---|---|---|
| GP | 48.36% | 99.93% | 100.00% |
| Transformer + DVA | 7.73% | 67.11% | 96.13% |
| CNN + DVA | 12.07% | 89.00% | 99.91% |

## H    PREDICTIVE UNCERTAINTY CALIBRATION RESULTS

## I    CONSOLIDATED HYPERPARAMETERS

To ensure reproducibility, we provide the detailed hyperparameters used for synthetic data generation, model architectures (Transformer, CNN, RNN, LSTM), and the optimization process. All models were implemented in `PyTorch`.

### SYNTHETIC DATA GENERATION (GP PRIOR)

For all experiments, training data was generated on-the-fly using Gaussian Process priors with Radial Basis Function (RBF) kernels. Output values $y$ were shifted to ensure positive support for the bucketization process. Table 11 lists the kernel parameters for each dimensional setting.

### MODEL ARCHITECTURE

Transformer models use standard Multi-Head Attention (or the proposed DVA modification) with Pre-LayerNorm. The architectural hyperparameters were tuned

CNN models map inputs to an embedding space and process them with 1D convolution blocks over the sequence dimension.

Recurrent models process the set as a sequence and then use an attention mechanism (VA or DVA) to aggregate context.

Table 11: Hyperparameters for Synthetic GP Prior Data Generation.

| Input Dim | Points / Dataset ($N$) | Kernel Type | Length-scale ($\ell$) | Kernel Var. | Noise ($\sigma^2$) | Output Shift |
|---|---|---|---|---|---|---|
| 1D | 100 | RBF | 0.6 | 0.01 | $1 \times 10^{-2}$ | 1.0 |
| 2D | 100 | RBF | 0.6 | 0.01 | $1 \times 10^{-2}$ | 1.0 |
| 5D | 400 | RBF | 0.6 | 0.001 | $1 \times 10^{-4}$ | 1.0 |
| 10D | 500 | RBF | 0.6 | 0.01 | $1 \times 10^{-4}$ | 1.0 |
| 64D (Power) | 500 | RBF | 215.0 | $1 \times 10^{-4}$ | $1 \times 10^{-4}$ | $U[0.9, 1.1]$ |

Table 12: Transformer Architecture Hyperparameters.

| Input Dim | Embed Dim | Encoder Layers | Heads | FFN Dim | Input Norm |
|---|---|---|---|---|---|
| 1D | 128 | 1 | 4 | 512 | Uniform |
| 2D | 128 | 1 | 4 | 512 | Uniform |
| 5D | 64 | 2 | 8 | 1024 | Uniform |
| 10D | 32 | 2 | 8 | 1024 | Uniform |
| 64D | 64 | 4 | 8 | 1024 | Standard |

Table 13: CNN Architecture Hyperparameters.

| Input Dim | Embed Dim | Layers | Kernel Size | Input Norm |
|---|---|---|---|---|
| 1D | 32 | 1 | 5 | Uniform |
| 2D | 32 | 1 | 5 | Uniform |
| 5D | 32 | 4 | 5 | Uniform |
| 10D | 32 | 4 | 5 | Uniform |
| 64D | 32 | 4 | 5 | Standard |

Table 14: Recurrent (RNN/LSTM) Architecture Hyperparameters.

| Input Dim | Embed Dim | Recurrent Layers | Attention Heads | Dropout |
|---|---|---|---|---|
| 1D | 64 | 1 | 4 | 0.1 |
| 10D | 64 | 4 | 8 | 0.1 |

Table 15: Optimization Hyperparameters.

| Setting | Epochs | Steps/Epoch | Batch Size | LR | Warmup Epochs |
|---|---|---|---|---|---|
| 1D (All) | 100 | 500 | 16 | $1 \times 10^{-3}$ | 25 |
| 2D (All) | 100 | 500 | 16 | $1 \times 10^{-3}$ | 25 |
| 5D (All) | 200 | 500 | 32 | $1 \times 10^{-3}$ | 50 |
| 10D (All) | 200 | 500 | 16 | $1 \times 10^{-3}$ | 50 |
| 64D (Power) | 200 | 500 | 32 | $1 \times 10^{-3}$ | 50 |

OPTIMIZATION AND TRAINING

All models were trained using the AdamW optimizer and a cosine-annealing learning rate schedule with linear warmup. The loss function used was the Bar Distribution (Riemannian) Negative Log-Likelihood.

## J SCALING TO FURTHER DIMENSIONS

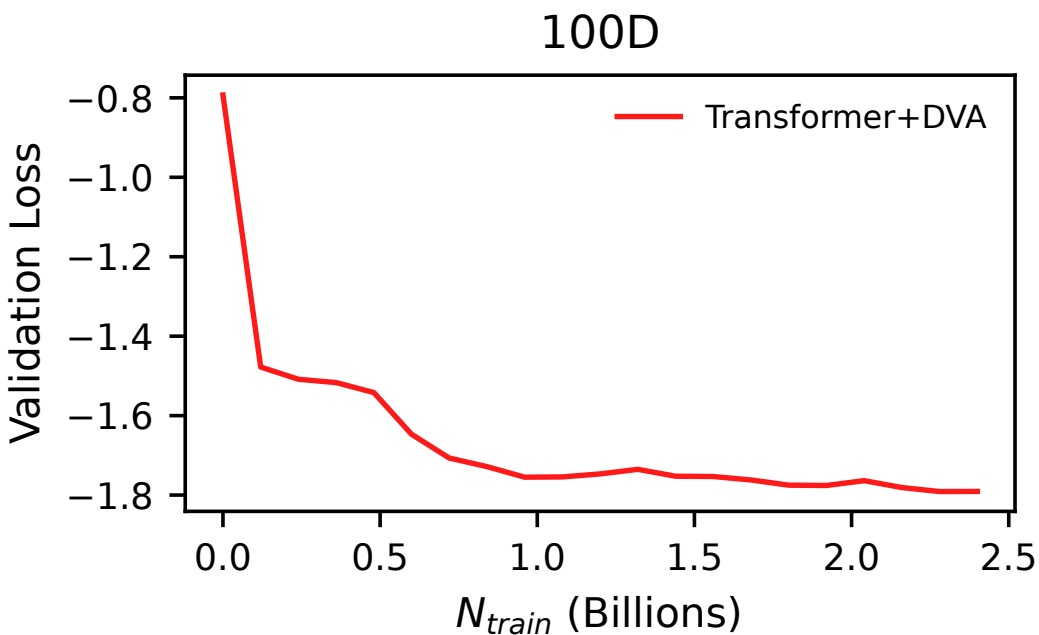

Figure 15: **Validation Loss on 100-Dimensional Data.** Performance of the Transformer model utilizing Decoupled Value Attention (DVA) trained over 200 epochs. The $x$-axis represents the cumulative number of training points observed (in billions). The model demonstrates stable convergence on the high-dimensional regression task. The $N_{context} = 1500$ and RBF prior was used with a fixed length scale. It took 81.5 hours on NVIDIA 4500ADA GPU having 24GB VRAM.

## K    EMPIRICAL COMPARISON WITH POST-HOC LOCALIZATION

To empirically validate our architectural approach against the post-hoc localization method proposed by Nagler (2023), we conducted a direct comparison. We applied two post-hoc localization strategies to a pre-trained VA PFN at inference time: (1) a **k-Nearest Neighbors (k-NN)** filter, where only the $k$ closest context points are used, and (2) an **Exponential** distance filter, where context points are selected based on a distance threshold controlled by a decay factor $\gamma$. The results, shown in Figure 16, provide two key insights.

- In the low-dimensional (1D) setting, where the VA model is able to learn a useful representation, post-hoc localization is effective in reducing baseline error, and even slightly outperforming DVA with certain neighborhood hyperparameters. This confirms the efficacy of the localization principle in low-dimensional regimes where the base model has learned a meaningful signal.

- In the high-dimensional (10D) setting, the VA PFN completely fails to learn from the data, with its MSE remaining high and flat regardless of context size. Consequently, applying post-hoc localization to this poorly trained model provides no benefit whatsoever; filtering an uninformative context set is ineffective. In contrast, DVA's architectural locality enables successful learning from the start, with its error decreasing consistently as more context is provided.

This experiment empirically validates our central hypothesis: post-hoc methods cannot rescue a model that has failed to learn a meaningful spatial representation during training. This underscores the necessity of an architectural solution like DVA for scaling PFNs to high-dimensional regression tasks.

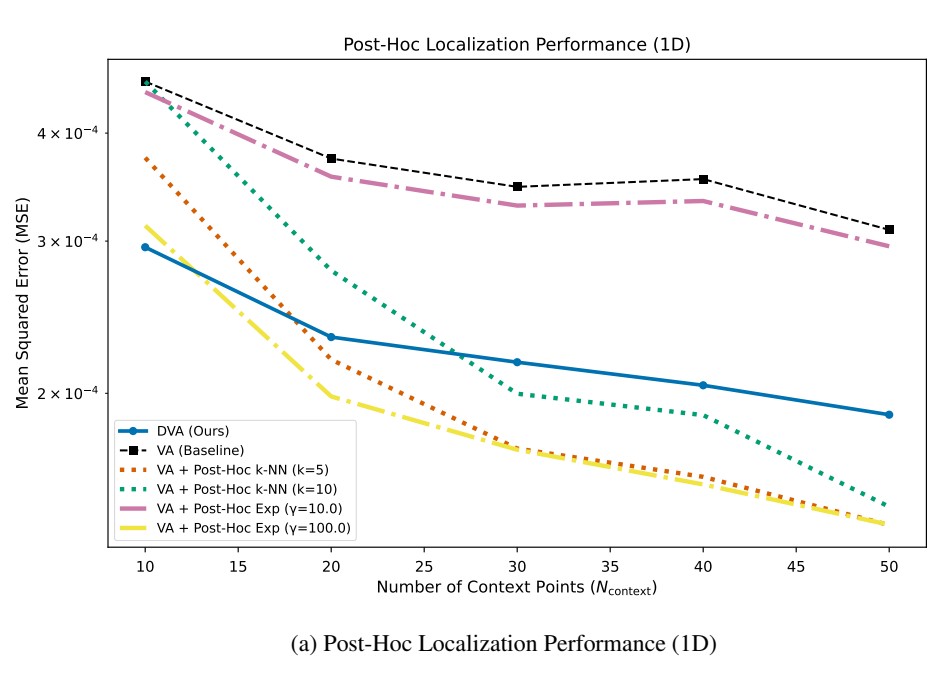

(a) Post-Hoc Localization Performance (1D)

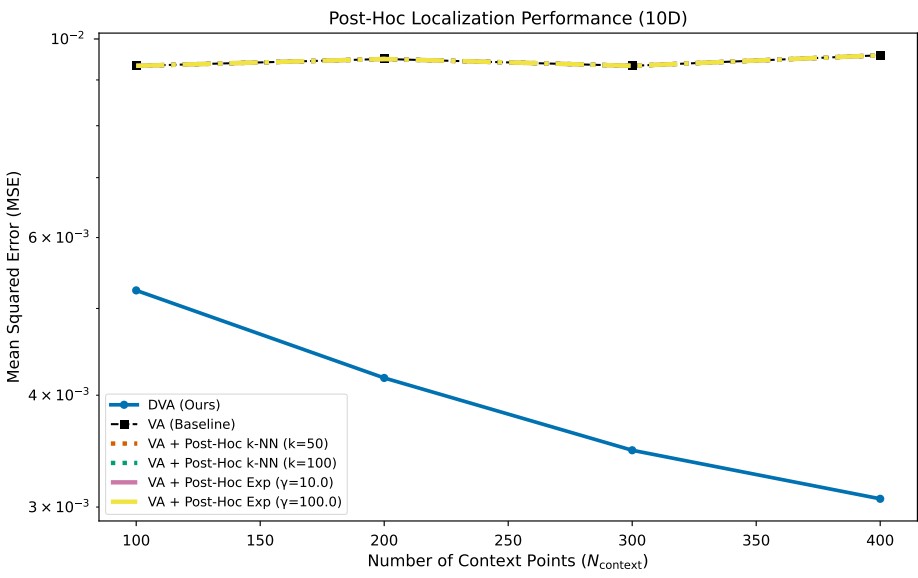

(b) Post-Hoc Localization Performance (10D)

Figure 16: Comparison of DVA with a baseline VA model augmented by post-hoc localization methods. In 1D, post-hoc methods are effective with correct choice of neighborhood parameters. However, in 10D, they fail to improve the poorly trained VA model, while DVA learns successfully.

### K.1 SENSITIVITY ANALYSIS OF POST-HOC K-NN LOCALIZATION

While post-hoc localization can improve upon a baseline VA model in 1D, its performance is critically dependent on the choice of the hyperparameter $k$ (the number of neighbors). To investigate this dependency, we conducted a simple sensitivity analysis in the 1D setting. We fixed the context size at $N_{\text{context}} = 30$ and evaluated the MSE of the VA + Post-Hoc k-NN model for $k \in [1, 40]$. We performed this experiment on a RBF prior with length-scale $0.6$.

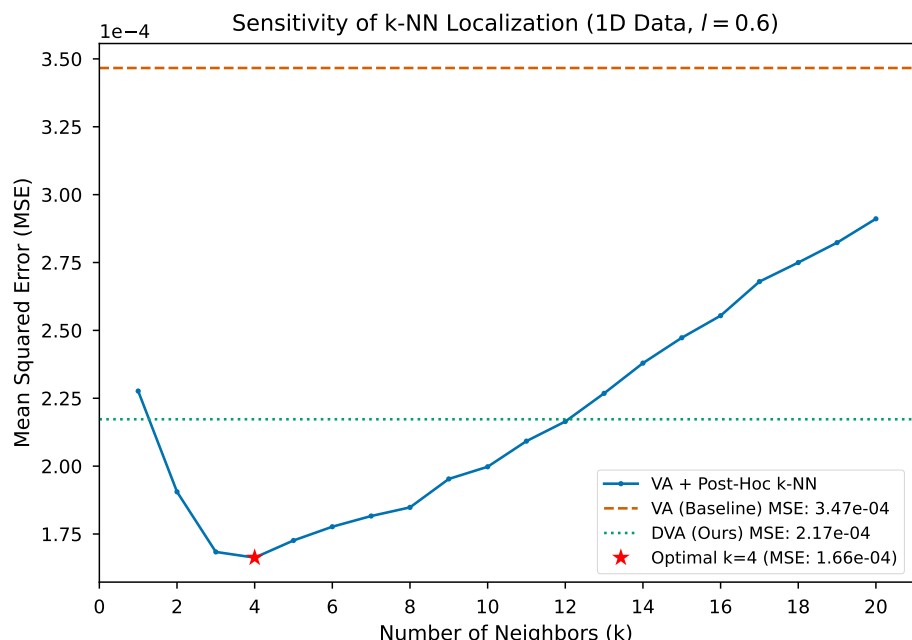

Figure 17: Sensitivity of Post-Hoc k-NN performance to the choice of $k$ on datasets with different smoothness levels. The optimal $k$ is data-dependent, highlighting the challenge of tuning post-hoc methods. DVA achieves robust, strong performance without requiring such tuning.

The results, shown in Figure 17, reveal a crucial challenge for post-hoc methods. The performance curve for k-NN exhibits a classic U-shape, representing a bias-variance trade-off: small $k$ leads to high variance (overfitting to noisy neighbors), while large $k$ leads to high bias (oversmoothing). Critically, the optimal value of $k$ is **data-dependent**. This demonstrates that post-hoc localization is not a "set-and-forget" solution. Achieving its optimal performance would require a new, costly hyperparameter search for $k$ for each new function class, undermining the inference-time efficiency of PFNs.

