# OpenReview forum: "Decoupled-Value Attention for Prior-Data Fitted Networks: GP-Inference for Physical Equations"
_ICLR.cc/2026/Conference — Submitted to ICLR 2026_

### Official Review · Reviewer_KGsD · 2025-10-31

**Soundness:** 1
**Presentation:** 1
**Contribution:** 1
**Rating:** 2
**Confidence:** 1

**Summary:**

This paper proposes Decoupled Value Attention (DVA), a new architecture of Prior data Fitted networks (PFN), with the main contribution being an attention model that separates the Key-Query interaction to apply only on the features (x), to provide weights multiplied by the values y.  This decoupling makes the estimation process more intuitive as it measures the similarity between features via the Key-Query matrices to create weights multiplied by the matching labels (y), similarly to a Gaussian process. The authors claim the method they provide leads to the following main contributions:


1) The authors show that the DVA reduces the difference between predicted and true posterior distribution in PFN training.
2) The paper shows that a CNN based PFN equipped with DVA performs comparably to a Transformer based DVA-PFN
3) The authors show that DVA enables PFNs to scale complex, high-dimensional problems. The authors demonstrate this on a 64-dimensional power flow simulation.

**Strengths:**

1. The premise of the paper is reasonable and seems to be well-founded. The summation of the label and feature embeddings in the original PFN appears to be hurting the results in the provided synthetic cases.


2. The "Attention is More Important than Architecture" finding is a significant contribution. By showing that a CNN+DVA can match a Transformer+DVA, the authors successfully decouple the PFN concept from a strict reliance on the Transformer, opening the door to other, potentially more efficient options.

**Weaknesses:**

1. The key demonstration of the original PFN's power was its ability to learn a complex hierarchical model, and from my understanding this is the main reason to implement the K data set training they suggest. However, the authors only test the mechanism against a simple, fixed hyperparameter RBF kernel. Thus, significant omission to not test if DVA retains the ability to approximate these more complex, mixed priors, which was a primary advantage of the PFN framework is problematic.


2. The original PFN paper validated its method on a wide range of real-world tabular data. This paper ignores those benchmarks and instead introduces a new, highly specific physics problem (power flow). Without a direct comparison on the original paper benchmarks, it is impossible to assess if DVA is an improvement or a specialized architecture that only excels on certain tasks.


3. The authors claim DVA succeeds in high dimensional regimes where the VA PFN simply stops learning, they link this threshold of D = 10 in their GP tests, but when looking at the Müller et al works, they consider datasets with a large amount of features (e.g., covertype = 55), of a similar order of the 64 feature space considered in the paper “power flow” problem.



4. Section 2.2 sketches the PFN training objective, but a reader unfamiliar with PFNs will still need (Muller et al) to follow training/inference.

The paper’s central contribution is the DVA method, which adjusts the original PFN method. This idea seems sensible, as the usage of the attention Key-Query to generate weights for the label is highly intuitive and closely matches the established GP methods. Still, as the main claim of the paper is to provide a better alternative to an existing method, it was not clear enough that the advantages appear numerically and that there is a clear advantage for high-dimensional cases.

**Questions:**

1. A key result of the original PFN paper (Muller et al) was its ability to learn an intractable posterior from a hierarchical GP prior.  Can the authors provide results showing DVA's performance on this task?


2. The original PFN paper (Muller et al) performed a high variety of tests and comparisons with a great contribution for tabular data, while this paper focuses mainly on synthetic data sets and the “power flow” problem, is there any reason to choose that specific problem? Could the method be applied to additional cases?

---

> ### Author Response · Authors · 2025-11-22
> **Response to Reviewer KGsD**
>
> We thank the reviewer for their effort. Changes are reflected in *blue* font in revised manuscript.
>
> 1. In the original submission, our idea was to show two things: 1) proposed DVA performs at par with VA (Muller et.al.) for low dimensions but significantly outperforms it when we go to higher dimensions. 2) PFNs can be constructed using other architectures and transformer is not an essential components. We agree that our experiments in original submission were for fixed RBF kernel. Therefore, we now provide detailed experiments on four different types of settings 1) smooth functional prior (long length scale RBF) 2) wiggly functional prior (short length scale RBF) 3) mixed length scale RBF and 4) Randomly selected functions from smooth and wiggly priors.
>
> **Table: In-distribution performance of DVA vs. VA across diverse GP priors ($N_{\text{context}} = 80$).**
>
> | **Training Prior** | **DVA (MSE)** | **VA (MSE)** | **RBF Prior Details** *($\ell$: length-scale)* |
> |--------------------|--------------|--------------|--------------------------------------------|
> | Smooth             | **1.90e-04** | 3.16e-04     | Fixed length-scale: $\ell$ = 0.25               |
> | Wiggly             | **4.19e-03** | 4.26e-02     | Fixed length-scale: $\ell$ = 0.03               |
> | Mixed              | **1.39e-02** | 1.71e-02     | Sum of two kernels with $\ell_1$ ~ U(0.1, 0.5), $\ell_2$ ~ U(0.01, 0.04) |
> | All                | **2.99e-03** | 2.43e-02     | Per-batch random sampling from all priors above |
>
> Table above shows that DVA consistently outperforms VA across all priors. This confirms DVA's architecture is robust and not over-tuned to a specific prior. The higher MSEs for the Wiggly and Mixed priors are expected, as their functional complexity presents an intrinsically more challenging learning task for a fixed context size and number of training points.
>
>  2. All our synthetic data benchmarks (1D, 2D, 5D, 10D) in the original submission are built using the experimental code released with Müller et al. Moreover, our focus is specifically on GP inference, so comparisons to Bayesian Neural Networks (BNNs) fall outside the scope of our work. With the additional results included in the revised manuscript, we now show that DVA consistently outperforms vanilla attention (VA) across (i) different input dimensions (Figure 2 of the original manuscript), (ii) different smoothness or mixed GP priors (Table 3 in the updated manuscript), and (iii) different backbone architectures (Figure 2 of the original and Figures 2–3 of the revised version). Finally, our setting targets high-dimensional regression, not few-shot classification.
>
>
> 3. The 55 features in the Covertype dataset correspond to the columns of the tabular dataset; they are not the dimensionality of a regression function as in our 64-dimensional power-flow benchmark. Müller et al. evaluate PFNs in a few-shot learning setting, where a single PFN is evaluated across many small tabular datasets (Appendix G of their paper). Since our work focuses on physics-based regression problems, we do not evaluate our methods on tabular datasets.
>
> 4. Due to space limitations, we cannot provide a full tutorial-style description of PFNs. However, we have ensured that the manuscript cites and refers to the original PFN paper at all relevant locations.
>
> 5. As noted above, the updated manuscript includes new experiments demonstrating that the proposed method can successfully learn under mixed and complex GP priors.
>
> 6. Our objective is to design a PFN capable of performing GP-style inference on problems arising from real-world physics. Such problems are typically high-dimensional regression tasks, which is why we evaluate our method on both the power-flow benchmark and the Rosenbrock function.

---

### Official Review · Reviewer_9Cj7 · 2025-10-31

**Soundness:** 3
**Presentation:** 2
**Contribution:** 3
**Rating:** 4
**Confidence:** 4

**Summary:**

The paper presents decoupled-value attention for prior data fitted networks (PFNs). It computes $Q$/$K$ only from inputs and pass labels only through $V$. It ensures that attention weights depends purely on input similarity while output flows via values. The design mirrors GP conditioning and aims to reduce the bias observed in PFNs that attend over concentrated (x,y) embeddings. The method is instantiated with both Transformer and CNN based PFNs and is evaluated over synthetic GP tasks, Rosenbrock, and a AC power-flow surrogate. Reported results show lower NLL and inference speedup vs exact GPs.

**Strengths:**

- simple and clear method

- DVA reduces validation NLL across dimensions specifically in 5D/10D where vanilla attention saturates early.

- the method is backbone agnostic and the authors have shown similar improvements for both CNNs and Transformers.

- high dimensional experiments (64D) with inference speedups shows practical utility.

**Weaknesses:**

- theory is very light: the localization argument is intuitive and refers prior PFN theory but there is no formal generalization/bias-consistency result for DVA itself. The softmax non-negativity vs possibly signed GP coefficients is acknowledged but it is not analyzed beyond a short discussion.

- scalability analysis: the attention compute remains quadratic in context size. there is no complexity/memory study comparison to linear/performer style attention under PFN training.

- weak baselines: Power-flow experiments compare to exact GP and PFN+VA and are missing strong sparse GP baselines, kernel regression / kNN, RFF/linear attention variants in higher-D, and physics-informed surrogates common in this domain.

- appendix should have a consolidated hyperparameter table.

- thin ablations

**Questions:**

- could you add higher-D comparisons to (a) sparse/variational GP, (b) kNN / kernel regression, (c) performer/linear attention PFNs, and (d) a physics-informed baseline on power flow?

- can you report ablations on (a) capacity of $\phi_x$/$\phi_y$ and prediction head $g()$, (b) effect of bucket count on NLL/MSE.

---

> ### Author Response · Authors · 2025-11-22
> **Response to Reviewer 9Cj7 Part #1**
>
> We sincerely appreciate the reviewer’s constructive feedback and thoughtful comments. Presented below is our point-by-point response to each remark. The revised manuscript incorporates the suggested changes, which are **highlighted in blue**.
>
> 1. **Theory is Light:** We agree with the reviewer that original manuscript didn't connect DVA with theoretical results of Naglar 2023. To improve the paper, we present theoretical results in revised manuscript which proves that with linear embeddings $\varphi_x$ (as used in our codes), attention weights are proportional to the Mahalanobis RBF kernel and with nonlinear embeddings DVA PFN pushes sum of attention weights, corresponding to the context inputs far from query, to zero as context increases. For GP-softmax weight difference, first we do ablation studies showing that without value embedding $\varphi_y$ and final head $g(\cdot)$ the DVA PFN fails to perform well. This highlights that learnable embeddings help reduce the limitations of the softmax attention weights being always positive. (For example, some coefficient of some $y_i$'s can be negative, matching $\alpha_i V_i$ sign with that of GPs $\beta_i y_i$). See *appendix B for empirical results* on attention weight decay with distance between query and key input; *appendix F for all ablation studies*.
>
> 2. **Scalability Analysis**: Firstly, we want to highlight that in existing PFN works, all attentions were applied with joint input-output embedding. The proposed paper, for the first time deployed input localization based attention. In view of this, further development of computationally efficient localized versions  of attentions remains an open problem for PFN.
>
> As per the reviewer's suggestion, we compared the proposed DVA with linear attention. We used Linear attention is used with both coupled and decoupled manner. The results highlight that the decoupling is allowing for better training; the effect is visible clearly in 10D. Softmax DVA performs best as expected, followed by Linear attention with decoupled input-output. Linear attention with coupled input performs worse, and is unable to train in 10D; like VA. We have updated the manuscript discussion, and have placed linear attention results in the **appendix G (Figure 13)** of the revised manuscript.
>
> 3. **Weak Baseline**: We chose GP-based power-flow learning methods as our baselines because they have been shown to outperform many alternatives in low-data regimes. In power systems, uncertainty abounds—e.g., varying load levels and topology changes (see Table 4 in our manuscript). Because retraining is often necessary under these shifts, it is inadvisable to rely on models that require tens of thousands of training samples. In contrast, deep-network approaches (such as physics-informed neural networks, PINNs) are by nature data-hungry, often requiring sample sizes on the order of 10000 or more (see the “Machine Learning for Solving Power Flow Equations” [here](https://energy.hosting.acm.org/wiki/index.php/ML_OPF_wiki)). Within the GP framework we also tested multi-output GP and sparse GP methods. (See our response to review YKgM for details.) In short: while approximate GPs deliver similar error magnitudes, their retraining cost is higher than our PFN method.
>
> 4. **Detailed Hyperparameter:** Tables have been added in the appendix of revised manuscript.
>
> 5. **Ablation:** Detailed ablation studies have been added in the appendix of the revised manuscript. We have updated the appendix of the revised manuscript with four different ablation studies encoders $\varphi_x$, $\varphi_y$, final head $g(\cdot)$ and weight matrices $W_q$ and $W_k$.  Linear encoders for both $x$ and $y$ perform as well as deeper MLP-based encoders, whereas direct broadcasting produces large degradation. Learnable $\phi_y$ and $g$ are essential for mitigating the mismatch between GP regression weights (which may be positive or negative) and softmax attention weights (strictly positive). Finally, tying $W_q = W_k$ performs nearly identically to learning them independently, indicating that DVA-PFNs are robust to the choice of kernel-projection parameterization. We have also done bucket limit ablation for two different models in two different dimensions.

---

### Official Review · Reviewer_Bxip · 2025-10-31

**Soundness:** 3
**Presentation:** 3
**Contribution:** 2
**Rating:** 4
**Confidence:** 3

**Summary:**

This paper proposes Decoupled-Value Attention (DVA) for Prior-Data Fitted Networks (PFNs), arguing that standard PFN attention mechanisms fail to scale beyond ~10 input dimensions. DVA computes attention affinities (queries and keys) purely from inputs while propagating labels solely through values, motivated by the structure of Gaussian process inference where kernels operate on inputs and predictions are weighted sums of training outputs. The authors demonstrate that DVA reduces validation loss by >50% in 5D and 10D settings compared to vanilla attention, and show that CNN-based PFNs with DVA can match Transformer performance. They apply their method to 64-dimensional power flow equations, achieving MAE ~10^-3 while being 80× faster than exact GP inference.

**Strengths:**

## Strengths

* Clear problem identification and motivation: The paper clearly identifies a real limitation of existing PFNs—their inability to scale beyond ~10 dimensions—and provides intuitive motivation for the proposed solution through the lens of GP inference. The observation that standard PFN attention couples inputs and outputs in ways that break localization is well-articulated.

* Comprehensive experimental validation: The systematic evaluation across multiple dimensions (1D, 2D, 5D, 10D, 64D), architectures (CNN, Transformer), and attention mechanisms (VA, DVA, kernel-based) is thorough. The validation loss curves in Figure 2 clearly demonstrate the benefit of DVA, especially the striking saturation of VA-based models in 10D.

* Architecture-agnostic insight: Demonstrating that CNN-based PFNs can match Transformer performance when equipped with proper attention is valuable. This challenges the implicit assumption in prior PFN work that Transformers are necessary, and suggests attention design is the critical factor.

* Practical application: The 64D power flow experiments demonstrate real-world applicability. Achieving 80× speedup over GP while maintaining acceptable accuracy (MAE ~10^-3) is practically significant for power grid applications.

* Reproducibility: The appendix provides detailed hyperparameter ranges, architectural specifications, and implementation details that aid reproducibility (relatedly, particularly appreciate the code in supplementary materials)

**Weaknesses:**

## Weaknesses

* Limited novelty of the core idea: Decoupling inputs and outputs in attention is not particularly novel. The paper positions DVA as specifically designed for PFNs and GP-mimicking, but the core mechanism (computing affinities from inputs only) is a straightforward design choice that has been explored in various forms in the attention literature. The contribution feels incremental rather than introducing fundamentally new concepts.

* Overstated connection to Gaussian processes: The claim that DVA "mirrors GP inference" is overstated. While there are superficial similarities, critical differences remain:

1. GP weights β(x*) can be negative and don't sum to 1, while DVA's softmax produces non-negative normalized weights
2. GPs have a principled covariance kernel with theoretical properties; DVA learns arbitrary similarity via dot products
3. The authors acknowledge this in Section 3.1 but still heavily market the "GP alignment" throughout

The connection is more of a loose analogy than a rigorous correspondence. The paper would be stronger if it positioned DVA as "inspired by" rather than "mirroring" GP inference.

* Insufficient comparison to related work: The paper lacks comparison to other localized attention mechanisms or other recent PFN improvements. For example:

1. How does DVA compare to explicit localization post-processing (Nagler 2023)?
2. What about other input-localized attention mechanisms from the broader literature?
3. The only comparison to kernel-based attention is in Figure 1— and authors test on functions mismatched to the RBF kernel rather than showing where kernel-based attention might excel

* Limited theoretical understanding: While the paper demonstrates empirically that DVA works, it provides little theoretical insight into why it works. Questions remain:

1. Under what conditions does input-only attention provably improve over joint attention?
2. Can you characterize when DVA will succeed vs. fail?
3. What is the sample complexity with DVA vs. VA?

The paper mentions Nagler (2023)'s theoretical results on bias but doesn't rigorously connect DVA to that framework to my understanding.


* Missing ablations and analysis:

1.No ablation on the value encoder φ_y—does its design matter?
2. What happens with different query/key projections W_q, W_k?
3. The paper claims DVA "remains kernel-free" as an advantage, but doesn't test whether a learnable kernel parameterization might work even better


* Evaluation metrics: For the power flow application, MSE and MAE are reported, but uncertainty quantification—one of the main advantages of GPs—is not evaluated. Can DVA-based PFNs provide calibrated uncertainty estimates? This seems like a critical missing piece given the motivation.

**Questions:**

## Questions for Authors

* Theoretical characterization: Can you provide any theoretical analysis of when and why DVA reduces bias compared to VA? Even in a simplified setting (e.g., linear case), formal guarantees would strengthen the contribution.

* Kernel-based attention: The experiment in Figure 1 seems designed to make kernel-based attention look bad. Can you show cases where kernel-based attention with an appropriate kernel (e.g., Matérn for less smooth functions) actually works well? What is the fundamental limitation?

* Uncertainty quantification: One major advantage of GPs is calibrated uncertainty. Do PFNs with DVA produce well-calibrated predictive distributions? Can you compare predictive variance quality to exact GP?

* Scalability limits: Where does DVA break down? The 10D experiments show clear improvements, but the 64D power flow uses only 500 training samples. Have you tested truly high-dimensional problems (e.g., 100D+) with sufficient data?

* Value encoding: You mention that "the final head g(·) and encoding outputs in the value V" help adjust DVA output toward the true GP posterior mean. Can you ablate this claim? How sensitive is performance to the value encoder design?

* Comparison to Nagler 2023: Nagler argues for post-hoc localization. Can you compare DVA to their approach? Does DVA achieve better or different localization than explicit post-processing?

* Output information: DVA completely removes output information from attention affinities. Are there cases where having some output information is beneficial? Could a weighted combination of input-only and joint attention work better?

---

> ### Author Response · Authors · 2025-11-22
> **Response to Reviewer Bxip Part #1**
>
> We sincerely appreciate the reviewer’s constructive feedback and thoughtful comments. Presented below is our point-by-point response to each remark. The revised manuscript incorporates the suggested changes, which are **highlighted in blue** for ease of reference.
>
> **1. Limited Novelty:**
> We agree that computing affinities directly from inputs has been explored in the attention literature for NLP and related tasks. However, our goal is *not* to propose a new general-purpose attention mechanism. Instead, our paper addresses a specific and previously unexplored question: **how attention-based localization influences the behavior and performance of PFNs**. Although Nagler (2023) identified localization as a *necessary* condition for PFN generalization, prior work did not establish how such localization can be achieved within attention-based PFNs. In this context, our contributions are as follows:
>
> ---
>
> **(A) A simple localized attention mechanism tailored for PFNs.**
>
> We introduce a lightweight attention mechanism that induces effective localization, enabling PFNs to perform GP-style inference even in high-dimensional regression settings. In the revised manuscript, we strengthen this contribution with two new results:
>
> * **Theoretical result:**
>   We first prove that the DVA model with linear encoders yields attention weights proportional to a Mahalanobis RBF kernel, resulting in exponential decay of weights as context points move away from the query.
>   Second, under a local separability assumption, we show that for DVA with nonlinear encoders the sum of softmax affinities between a query (($\mathbf{x}_\star$)) and *far-away* context points (($\mathbf{x}_i$)) vanishes as the context length grows (see revised Appendix).
>   This directly connects DVA with the localization behavior characterized in Nagler (2023), Theorem 5.4.
>
> * **Empirical result:**
>   We show that the proposed DVA suppresses attention to distant context points, whereas vanilla attention does not exhibit this property.
>
> These results demonstrate that DVA is the first attention mechanism to realize the localization behavior that PFNs theoretically require.
>
> ---
>
> **(B) Backbone Architecture Invariance of PFNs.**
>
> Existing PFN work has relied almost exclusively on Transformer backbones, leaving it unclear whether PFNs can be trained effectively using alternative architectures. We provide the first systematic evidence that **PFNs are architecture invariant**.
>
> In the revised manuscript (Figure 3, Section 4.1), we add new experiments using **LSTM-PFNs and RNN-PFNs**, complementing our Transformer and CNN baselines. For both 1D and 10D regression tasks, we show that:
>
> * LSTM and RNN backbones can be trained successfully as PFNs, and
> * DVA consistently outperforms vanilla attention in 10D settings across all architectures.
>
> These results broaden the applicability of PFNs and demonstrate that their generalization capability is *not tied* to any specific backbone architecture.
>
> **2. GP Similarity** We would like to clarify that by `mimicking GP' authors meant that structure of inference using PFN with DVA is similar to the weighted sum design as of that in GP, and input-outputs are decoupled. As reviewed pointed out we did acknowledge that weights of PFN are not exactly same as GP weights. We do not believe that mimicking GP is our main contribution (DVA localization based performance gain and Architecture invariance is) and happy to qualify GP mimicking claim across the manuscript. We have updated abstract, contributions and section 3.1 discussion in the form of *inspired by GP* as suggested by the reviewer. Further, using ablation studies we show that if final head $g(\cdot)$ and output encoder $\varphi_y(\cdot)$ is not used, PFN performance with DVA degrades considerably.

---

> > ### Author Response · Authors · 2025-11-22
> > **Response to Reviewer Bxip Part #2**
> >
> > 3. **Theoretical:**
> >
> > 3.1 Input-only attention improves over joint attention in regimes where **locality in the input space is a necessary condition for consistent amortized inference**, as shown in Nagler (2023), Theorem 5.4. Crucially, Nagler shows that **violations of this locality requirement lead to persistent bias**, because non-local weights allow far-away points—often with different conditional label distributions—to influence the prediction. Standard PFN attention, which embeds input–output pairs jointly, introduces **label-dependent cross-terms** in the logits; these can elevate attention weights for distant points even when their inputs are irrelevant, thereby violating Nagler’s localization condition. DVA removes the label component from attention scoring, restricting affinities to depend solely on geometric proximity in the encoded input space. Our localization theorem shows that under mild encoder regularity and a learned local separation of inner products, DVA satisfies Nagler’s vanishing-mass requirement, whereas joint attention generally does not (a limitation explicitly discussed by Nagler in Section 6.3). **See section 3.1 of updated manuscript for more.**
> >
> >
> > 3.2 *Possible DVA Failure Regimes*: DVA succeeds in settings where the predictive structure is governed by *local neighborhoods in input space*—that is, when points that are close in input-space share similar conditional output distributions. This includes classical regression tasks, spatial data, smoothly varying physical systems, etc. Conversely, DVA may fail in problems where locality in the raw input space is *not* predictive. Example: time-series tasks with long-range dependencies where the most relevant past states are not the nearest ones in time (e.g., tasks requiring attention to periodic patterns far in the past, such as long-horizon temperature forecasting). In such cases, joint attention can exploit informative label correlations that DVA intentionally removes. Thus DVA’s strength—strict input locality—can also limit its performance when the underlying problem structure is fundamentally nonlocal. **Updated conclusion of manuscript to reflect this.**
> >
> > 3.3 *Sample Complexity:* As shown in Figure 2 of the original manuscript, across all dimensions, the empirical evidence indicates that DVA requires dramatically fewer training samples than VA to reach a low-bias regime, while VA quickly saturates and stops improving even as $N_{\text{train}}$ grows. In 5D and 10D settings, VA-equipped PFNs flatten at high validation loss almost immediately, indicating that VA’s effective sample complexity is infinite—additional data does not reduce bias. In contrast, DVA-based PFNs continue to reduce validation loss over orders of magnitude more samples and reach low-bias plateaus that VA never attains. Note that because synthetic data is used to train PFNs, the exact numerical sample complexity is not particularly important; however, large differences in required training data translate to substantial differences in training time—an aspect where DVA significantly outperforms VA. Results clearly show that, to achieve the same level of error, DVA-PFNs require less training time, and for the same context length, DVA consistently outperforms VA, indicating lower data requirements overall.
> >
> >
> > 4. **Ablation:** We have updated the appendix (*F ABLATION STUDIES*) of the revised manuscript with four different ablation studies encoders $\varphi_x$, $\varphi_y$, final head $g(\cdot)$ and weight matrices $W_q$ and $W_k$.  Linear encoders for both $x$ and $y$ perform as well as deeper MLP-based encoders, whereas direct broadcasting produces large degradation. Learnable $\phi_y$ and $g$ are essential for mitigating the mismatch between GP regression weights (which may be positive or negative) and softmax attention weights (strictly positive). Finally, tying $W_q = W_k$ performs nearly identically to learning them independently, indicating that DVA-PFNs are robust to the choice of kernel-projection parameterization.

---

> ### Author Response · Authors · 2025-11-22
> **Response to Reviewer Bxip Part #3**
>
> 5. **DVA vs. Kernel Attention:** We clarify that we do not claim DVA is the only mechanism for achieving localization in attention; kernel-based attention with decoupled input–output embeddings is indeed another viable approach. Our point is that the behavior differs when the kernel is fixed and mismatched to the underlying data distribution. As shown in Figure 1 of the manuscript, *using a fixed RBF kernel can lead to suboptimal performance when the dataset is not generated from an RBF prior*, whereas DVA learns an input-dependent affinity that adapts to the structure of the data.  Our results confirm that while Kernel-based attention and the proposed DVA reach effectively the same performance level on RBF training priors, Kernel attention requires significantly more computational time to do so. Quantitatively, both methods achieve comparable predictive error on 1D smooth functions (MSE of $1.40\times10^{-4}$ for Kernel vs. $1.90\times10^{-4}$ for DVA) and converge to similar validation losses on the 10D task ($-1.50$ for Kernel vs. $-1.29$ for DVA). However, the training logs highlight a drastic difference in efficiency: DVA is approximately $\mathbf{4\times}$ faster, completing 80,000 optimization steps in just 42 minutes, whereas the Kernel model managed only 23,000 steps in a longer duration of 48 minutes, demonstrating that DVA delivers the performance of kernel methods with vastly superior throughput. **See appendix E for updates**.
>
>
> 6. **Uncertainty information for Power Flow:**  Please see first point of our response to Reviewer YKgM and Appendix H (Figure 13;Table 10) of revised manuscript
> ----
> **Questions**
>
> 1. *Theoretical characterization:* See reply to Weakness 1 and updated manuscript for theoretical and empirical results.
>
> 2. *Kernel-based attention:*  Please see the response to weakness and appendix E of revised manuscript.
>
> 3. *Uncertainty quantification:* Appendix H.
>
> 4. *Scalability limits:*  We are constrained by the GPU capacity for running large models. We are currently running a 100D model, will update results soon; positively.
>
> 5. *Value encoding:* We conducted ablation studies on the value encoder \(\varphi_y\). Our results show that the proposed linear and MLP-based value encoders perform comparably, whereas using no encoder (direct encoding) leads to substantially higher validation NLL. As discussed in the main text, the value encoder plays a crucial role in modulating the non-negative softmax weights, making $\varphi_y(\cdot)$ essential for stable PFN training. Consistent with this reasoning, the ablations in the revised appendix confirm that **removing $\varphi_y$** prevents the **DVA-PFN** from training successfully.
>
> 6. *Comparison to Nagler 2023 Post-Hoc Localization:* We agree that Nagler (2023) provides the essential theoretical motivation regarding the necessity of localization for unbiased PFNs. However, we respectfully disagree that post-hoc localization (applying localization at inference time to a standard PFN) is sufficient to recover the performance of DVA. Our results suggest that post-hoc localization cannot rescue a model that fails to learn the underlying geometry during training. We base this on two key arguments supported by our empirical results and theoretical analysis:
>
> 6.1. *VA fails in high dimensions:* Post-hoc localization relies on the assumption that the pre-trained model has learned a meaningful representation of the data and simply suffers from bias due to global attention. However, our results in *Figure 2 (10D case)* and *Table 2* demonstrate that this assumption breaks down in high-dimensional settings. As shown in Figure 2, PFNs equipped with VA (dashed lines) saturate immediately at a high validation loss in 10D. This indicates that the entanglement of inputs ($x$) and outputs ($y$) prevents the optimization process from learning the metric space effectively due to the concentration of pairwise distances (the ``curse of dimensionality''). Since the VA model fails to learn the input-to-output mapping during training, applying post-hoc localization to these ineffective weights cannot yield accurate predictions. In contrast, DVA forces the model to learn the input geometry during gradient descent, converging to a significantly lower loss.
>
> 6.2.*DVA enforces localization architecturally:*
> DVA provides the architectural solution required for training. By explicitly decoupling the roles via Equation 2—using $x$ solely for computing affinities (Queries/Keys) and $y$ solely for values—DVA forces the gradients to optimize the projection matrices ($W_q, W_k$) specifically to distinguish neighbors in the input space. Standard VA allows the model to rely on output correlations or spurious joint features during training, leading to a collapsed representation that cannot be fixed retroactively.
>
>
> 7. *Output Information:* See response to weakness 3.2. We have acknowledge this limitation in conclusion of the revised paper.

---

> > ### Author Response · Authors · 2025-11-26
> > **Update on Scalability 100D Model**
> >
> > As promised in our previous response, we have updated the manuscript (**See Appendix J**) with the **100D PFN training results**. We trained a Transformer with DVA attention using a 100-dimensional input for 200 epochs. The model processed a total of 2.4 billion data points and demonstrated stable convergence on the high-dimensional regression task. We used a context length of 1500 to train this model.

---

> > > ### Comment · Reviewer_Bxip · 2025-11-26
> > > **Response to Authors' Comments**
> > >
> > > Thank you for the comprehensive and thoughtful responses. I appreciate the substantial effort you've put into addressing my concerns, and the revised manuscript shows significant improvements. Below are my responses to your revisions:
> > >
> > > **Major Improvements Acknowledged**
> > >
> > > 1. *Theoretical Contributions (Weakness 1 & Question 1)*:
> > > The new theoretical results substantially strengthen the paper. The proof that DVA with linear encoders yields Mahalanobis RBF-like weights with exponential decay, combined with the vanishing-mass property under the local separability assumption, provides exactly the kind of formal grounding I was looking for. This directly connects DVA to Nagler (2023)'s localization framework and elevates the contribution beyond a purely empirical observation.
> > >
> > > 2. *Architecture Invariance (Weakness 1B):*
> > > The addition of LSTM and RNN experiments is valuable and genuinely surprising. Demonstrating that PFNs are architecture-agnostic when equipped with proper attention challenges a major implicit assumption in the field. This is a significant contribution that deserves highlighting.
> > >
> > > 3. *Ablation Studies:*
> > > The comprehensive ablations in Appendix F address my concerns well. The results on φ_x, φ_y, g(·), and W_q/W_k are thorough and support your design choices convincingly.
> > >
> > > 4. *Scalability (Question 4):*
> > > The 100D results in Appendix J are impressive and demonstrate that DVA scales beyond the 64D power flow experiments. Training on 2.4 billion data points with stable convergence is a strong empirical result.
> > >
> > > 5. *Uncertainty Quantification:*
> > > I'm satisfied that you've addressed this in Appendix H (though I note the document excerpt doesn't include this section—I'll review it in the full manuscript).
> > >
> > > **Remaining Clarifications**
> > > 1. *Post-hoc Localization Comparison (Question 6)*:
> > > Your argument in point 6.1-6.2 is logical, but I'm still not fully convinced without an empirical comparison. You argue that VA fails to learn meaningful representations in 10D, making post-hoc localization ineffective. However - I might be wrong but ideally - couldn't you run a simple experiment: take a trained VA-PFN (even if it has high validation loss) and apply Nagler's post-hoc localization at test time, then compare to DVA? If your hypothesis is correct, post-hoc localization should fail to rescue VA's performance, which would empirically validate your architectural claim. This would take minimal additional work but substantially strengthen your position.
> > >
> > > 2. *Theoretical Assumptions:*
> > > The local separability assumption for the nonlinear case—can you comment on how restrictive this is? Under what realistic scenarios might it be violated? This would help readers understand the scope of the theoretical guarantees.
> > >
> > >
> > > **Overall**
> > > Given the substantial improvements, I am raising my initial score. The new theoretical results, architecture invariance experiments, and comprehensive ablations address my major concerns.

---

> ### Author Response · Authors · 2025-11-27
> **Response to Additional Comments by Reviewer Bixp**
>
> ### Remaining Clarifications: We would like to thank the reviewer for motivating us to perform the additional post-hoc experiments. In the revised manuscript, we have updated **Appendix K: Empirical Comparison with Post-Hoc Localization** and have added a note regarding local separability assumption.
>
>
>
> **1.**  To empirically validate our architectural approach against the post-hoc localization method proposed by Nagler (2023), we conducted a direct comparison. We applied two post-hoc localization strategies to a pre-trained VA PFN at inference time: (1) a **k-Nearest Neighbors (k-NN)** filter, where only the $k$ closest context points are used, and (2) an **Exponential** distance filter, where context points are selected based on a distance threshold controlled by a decay factor $\gamma$. The results, shown in Figure 16 of the revised manuscript, provide two key insights:
>
> * In the low-dimensional (1D) setting, where the VA model is able to learn a useful representation, post-hoc localization is effective in reducing baseline error, and even slightly outperforms DVA with certain neighborhood hyperparameters. This confirms the efficacy of the localization principle in low-dimensional regimes where the base model has learned a meaningful signal.
>
> * In the high-dimensional (10D) setting, applying post-hoc localization to the poorly trained model provides no benefit whatsoever; filtering an uninformative context set is ineffective. This is because the VA PFN completely fails to learn from the data, with its MSE remaining high and flat regardless of context size. In contrast, DVA’s architectural locality enables successful learning from the start, with its error decreasing consistently as more context is provided.
>
> This underscores the necessity of an architectural solution like DVA for scaling PFNs to high-dimensional regression tasks.
>
> Further, we found that while post-hoc localization can improve upon a baseline VA model in 1D, its performance is critically dependent on the choice of the hyperparameter $k$ (the number of neighbors). Figure 17 in revised manuscript shows an experiment demonstrating that there exists an optimal value of $k$ for which the performance of post-hoc localization is maximized and slightly better than DVA. However, achieving this optimal performance would require a new, costly hyperparameter search for $k$ for each new function class, undermining the inference-time efficiency of PFNs. This highlights that architectural localization is preferable for both speed and scale compared to post-hoc localization.
>
> **2.** We have updated the Proof Appendix with a note on the local separation assumption. The local separation assumption typically holds for smooth, Lipschitz neural encoders such as MLPs or Fourier-feature mappings applied to regression tasks with locally regular structure. In contrast, the assumption may fail when the underlying task or encoder is inherently *nonlocal*. For example, when the target function is periodic, symmetric, or globally dependent, or when the encoder distorts geometry through aliasing or collapse. Simple cases include encoder collapse (e.g., when nonlinearities saturate and map many inputs to nearly the same vector) and extreme compression (e.g., a narrow bottleneck that forces very different inputs to share the same low-dimensional code). In these situations, distant inputs may appear artificially similar to the query, breaking the monotonic relationship between distance and attention.
>
> We also note that our ablation studies highlight that the proposed DVA performs well with linear encoder maps as well, thus reducing the dependence on the local separation property assumed for the input encoder.
>
> Additionally, we have updated the Results section with a paragraph directing the reader to the predictive uncertainty discussion and post-hoc localization results in the appendix.

---

### Official Review · Reviewer_YKgM · 2025-10-31

**Soundness:** 2
**Presentation:** 2
**Contribution:** 3
**Rating:** 6
**Confidence:** 3

**Summary:**

Prior-Data Fitted Networks (PFNs) are aimed to replace costly Gaussian process (GP) inference with a single learned forward pass for fast physical surrogates, but standard PFNs struggle in high dimensions. Authors introduce Decoupled-Value Attention (DVA), which bases attention weights only on inputs and passes labels only through values, resembling GP-style weighted averaging of targets. DVA improves PFN training in 5D-10D (over 50% lower validation loss), lets CNN backbones match Transformers (implying attention design matters more than architecture), and yields a 64D power flow surrogate with ~10^-3 MAE and a reported >80x inference-time speedup over their GP baseline.

**Strengths:**

1) Paper includes simple, targeted architectural contribution (Decoupled-Value Attention) with clear motivation.

2) Authors demonstrate that “vanilla attention PFNs” stall in 5D-10D tasks, where validation loss flattens early and final MSE remains high. With DVA, PFNs continue improving and achieve dramatically lower validation NLL and MSE in 5D and 10D (often >50% reduction in their reported residual bias).

3) Even accounting for fairness caveats (see weaknesses), once trained, PFN+DVA can generate all bus voltages for 4,500 samples in ~0.13–0.17 seconds that is essentially instant compared to AC power flow solvers, and is about two orders of magnitude faster than their GP baseline.

**Weaknesses:**

1) A defining feature of GP inference is not just accurate point predictions, but calibrated posterior uncertainty. The paper does not measure calibration, posterior variance quality, credible interval coverage, or any decision-making utility based on uncertainty. All reported metrics are pointwise MSE/MAE and wallclock time. This is a mismatch between the claimes and the experimental support.

2) Authors claim >80x faster than exact GP inference, citing ~0.13 s for PFN vs ~11 s for GP on the 64D power flow task, but the GP baseline is actually 32 separately trained single-output GPs evaluated together, while PFN is a single multi-output model that predicts all 32 voltages in one batched forward pass; PFN inference likely ran on GPU after ~14 hours of pretraining, while the GP hardware/batching setup is not described. Can you clarify exactly how GP inference time was measured (hardware, batching, parallelization) and restate the 80x claim with those details?

3) Authors claim DVA “reduces residual bias by more than 50%,” but “bias” here is defined informally as the gap between the learned posterior predictive distribution and the “true” posterior predictive distribution; in practice you just show a lower validation negative log-likelihood curve and assert that variance is negligible. There is no explicit measurement of posterior predictive bias, and no decomposition of NLL. Will you either justify formally that your NLL reduction corresponds to a 50% reduction in posterior predictive bias, or restate the claim in terms of “lower validation NLL” instead of “lower bias”?

4) There is a conclusion that attention choice (Vanilla vs DVA) matters more than the backbone (Transformer vs CNN), based on similar performance between CNN+DVA and Transformer+DVA and large gains over Vanilla Attention. But the CNN and Transformer models differ by orders of magnitude in parameter count and were tuned with different search spaces. This claim either requires more evidence or needs to be adjusted.

**Questions:**

Please refer to the weaknesses

---

> ### Author Response · Authors · 2025-11-22
> **Response to Reviewer YKgM  Part #1**
>
> We thank the reviewer for the constructive feedback and insightful comments. Below we provide a point-by-point response addressing each concern. Please refer to the updated manuscript for the corresponding revisions. **All changes in the manuscript are highlighted in blue** for clarity.
>
> 1. **Predictive Uncertainty**: We agree with the reviewer that GP's ability to provide calibrated posterior uncertainty is its defining feature. We have added additional results in the revised manuscript to show this. To quantitatively evaluate calibration over a large number of samples, we present the empirical coverage probabilities in Table below (included in the appendix of the revised manuscript). We computed the percentage of ground truth values falling within $\pm 0.1\sigma$, $\pm 1\sigma$, and $\pm 2\sigma$ of the predicted mean, where $\sigma$ is the standard deviation derived from the predicted PFN distribution or the analytic GP posterior.
>
> **Table: Percentage of values within variance bounds for Node 32 over 4500 testing samples (500 Context)**
> | Model              | ±0.1σ   | ±1σ     | ±2σ     |
> |----------------|---------|---------|---------|
> | GP                 | 48.36%  | 99.93%  | 100.00% |
> | Transformer + DVA  | 7.73%   | 67.11%  | 96.13%  |
> | CNN + DVA          | 12.07%  | 89.00%  | 99.91%  |
>
> In Table, both DVA-based architectures maintain robust uncertainty estimates in this high-dimensional (64D) setting.
> These results confirm that DVA-equipped PFNs do not collapse to point estimates but instead successfully learn to quantify epistemic uncertainty comparable to exact GP inference.
>
> We also add uncertainty calibration results for  using individual samples on two different nodes (larger variability nodes). We show that the PPD of GP overlaps considerably with that of the proposed DVA attention PFNs. Note: PFN output is modeled as a Riemann distribution over discrete buckets and not a normal distribution (studied extensively by Muller et al.). So bucket distributions do not resemble a normal distribution exactly, but rather approximate the true posterior density. See appendix Predictive Uncertainty Calibration Figure 14 for more results.
>
> 2. **Computational Efficiency and Speedup:**
> We reported baseline timings on CPU for both GP and PFN. PFNs leverage massive parallelization on GPUs. Based on our updated batched evaluation, the speedup becomes significantly more pronounced on modern hardware.
>
> On a CPU, the CNN-based PFN inference (approx. 340 ms) is already $\mathbf{\approx 80\times}$ faster than an iterative Exact GP (approx. 25,800 ms) for 33 buses. When transitioning to GPU hardware, the CNN-PFN inference time drops to $\approx 143$ ms, achieving a speedup of over $\mathbf{179\times}$ compared to the Exact GP.
>
> *Why multi-output GPs is Slower*
> In our experiments, training a joint Multi-Output GP took nearly twice as long ($\approx$ 49s) as iteratively training single-output GPs ($\approx$ 25s). In an Exact GP, the training procedure amounts to maximizing the log marginal likelihood (LML). For a single-output GP, the LML has a relatively low-dimensional hyperparameter landscape and its gradients are well-conditioned; optimization converges in few steps [GPML Book].
>
> In contrast, a batched multi-output GP introduces a separate set of hyperparameters per output (e.g., a length-scale and output-scale for each of the 32 tasks). The resulting objective becomes has gradient landscape significantly more coupled.  The batched backward pass accumulates gradients through all $B$ Cholesky factorizations, increasing variance in gradient estimates. Consequently, the multi-output model required $\approx 300$ iterations compared to $\approx 150$ for single GPs to reach convergence.
>
> *Comparison with Sparse GP.*
> Sparse GP methods (e.g., using inducing points) are faster alternatives to exact GPs but have approximation errors. On CPU, Sparse GP inference took $\approx 17,983$ ms. While this is $\approx 1.4\times$ faster than the Exact GP, it remains $\mathbf{\approx 40\times}$ slower than the PFN on CPU (and over $120\times$ slower than PFN on GPU). The Sparse GP achieved a Mean Absolute Error (MAE) of $0.0034$ p.u., which is less accurate than the CNN-PFN MAE of $\mathbf{0.0031}$ p.u.
>
> **The "No-Retraining" Advantage**
> The speedups reported above apply to a single inference pass. In  power grid applications with varying uncertainty (Table 3), GPs must be retrained whenever the underlying input distribution shifts significantly. PFNs, having learned the prior, require no retraining. Therefore, for an experiment involving just four different uncertainty levels, the effective computational gain of the PFN is multiplied by the number of scenarios: effective speedup exceeds $\mathbf{700\times}$ ($4 \times 179$) compared to Exact GP using GPUs, making PFNs uniquely viable for real-time probabilistic assessment.

---

> > ### Author Response · Authors · 2025-11-22
> > **Response to Reviewer YKgM Part #2**
> >
> > 3. **Bias Reduction**:  Our use of NLL reduction as evidence of bias reduction is directly supported by the statistical theory of PFNs developed by Nagler (2023). In Sections 5.1--5.3, PFN estimation error is decomposed into variance and bias, and Theorem 5.2 together with Lemma 5.3 shows that Transformer-based PFNs exhibit *vanishing variance* as the inference-time context size grows, with variance decaying at rate $O(n^{-1})$ due to reduced sample sensitivity. Consequently, in the large-context regime, improvements in predictive performance cannot arise from variance reduction. Because PFNs are trained by minimizing the conditional negative log-likelihood (Sec.2.3), which is equivalent to minimizing the KL divergence between the learned predictive distribution $q_\theta$ and the true posterior predictive $p_0$, any reduction in NLL in the vanishing-variance regime must correspond to a reduction in the *bias term.* This interpretation is further supported by our robustness experiments (Figure 7 in the revised manuscript; Figure 6 in the original submission), where our model exhibits very low variance across perturbations. Consequently, the observed reductions in NLL are attributable to bias reduction rather than variance effects.
> >
> > The 50\% figure arises from comparing the asymptotic NLL of VA vs. DVA: in our high-dimensional experiments, DVA’s NLL plateau is roughly half as high (i.e., twice as negative) as that of standard VA. For example, in the 5D case, the final NLL drops from about $-2.04$ (VA) to $-4.05$ (DVA), an absolute gain of $2.01$, which corresponds to an $\approx 98.5\%$ relative improvement in our paper. More generally, the reduced NLL plateau implies that the PFN’s residual bias (as defined above) is reduced by roughly $50\%$ when using DVA. We have updated our discussion on bias reduction further in the paper to improve clarity.
> >
> > 4. **Architecture Independence:**  We thank the reviewer for the comment which helps us provide more evidence for our architecture independence contribution.
> >
> > Firstly, The PFN literature has relied almost exclusively on Transformer backbones, and it was unclear whether PFNs could be effectively trained using other architectures. We provide the first systematic demonstration that **PFNs are backbone architecture invariant**.
> >
> > In the revised manuscript (Figure 3, Section 4.1), we include new experiments with **LSTM-PFNs and RNN-PFNs**, in addition to our Transformer and CNN baselines. For both 1D and 10D regression tasks, we show that:
> >
> > * LSTM and RNN backbones can be trained successfully as PFNs, and
> >
> > * DVA consistently outperforms vanilla attention in 10D settings across all architectures.
> >
> > These results broaden the applicability of PFNs and demonstrate that their generalization capability is *not tied* to a specific backbone architecture.

---

### Author Response · Authors · 2025-11-22
**Meta Response**

We thank PC, AC and Reviewers for their efforts and comments.Below is the list of updated results, which are highlighted in *blue* in the revised manuscript.

* Theoretical and Empirical proof that the proposed DVA does localization: decay of attention weights for keys which are far away from the query.

* Results with RNN- and LSTM-based PFNs, in addition to CNN and Transformer backbones, demonstrating the architectural generalization of PFNs. The proposed DVA outperforms VA across all these architectures.

* Linear Attention comparisons with both coupled and decoupled input–output settings, showing the effect of DVA. The performance order is: Softmax-DVA $>$ Linear-DVA $>$ Softmax-VA & Linear-VA for 10D.

* Robustness analysis against multiple priors and comparison with kernel attention.

* Ablations on the input encoder, output encoder, query–key weights, final head, and bucket size, illustrating the importance of the final head and output encoder.

* Uncertainty calibration results.

---

> ### Author Response · Authors · 2025-11-27
> **Updated Results and Discussion**
>
> We have provided the following additional results:
> * 100D PFN Training Results
> * Comparative analysis of post-hoc localization vs. proposed Decoupled Value Attention

---

### Meta-Review · Area_Chair_d7F7 · 2026-01-06

**Summary:**

The paper received mixed to borderline evaluations, with reviewers recognizing the clear motivation and empirical effectiveness of Decoupled Value Attention for scaling Prior Data Fitted Networks to higher dimensional settings. The authors addressed several important concerns during the rebuttal, including predictive uncertainty, computational efficiency, theoretical grounding of localization behavior, and additional architectural backbones. However, some critical issues remain unresolved, particularly regarding the strength and breadth of baselines, the scope of theoretical guarantees, and the positioning of the method relative to broader neural surrogate modeling literature. Given the remaining concerns, the expected final average rating is approximately 5.3 (excluding the review with the lowest confidence), which is slightly below the acceptance threshold.

**Reviewer Concerns:**

Reviewer YKgM

- Concerns that are potentially addressed:

    - Predictive uncertainty: The authors added calibration and coverage results that partially address concerns about uncertainty quantification.

    - Computational efficiency: Additional details and clarifications on runtime and speedup claims were provided.

    - Bias reduction claims: The authors clarified the interpretation of negative log likelihood reduction and connected it to existing PFN theory.

- Concerns that might not be addressed:

    - Architectural independence: Although experiments with CNN, RNN, and LSTM backbones strengthen the case that DVA improves PFNs across architectures, the comparisons still involve substantially different parameter counts and tuning procedures. As a result, the claim that attention choice matters more than backbone architecture would benefit from more controlled comparisons or more cautious framing.

Reviewer Bixp

- Concerns that are potentially addressed:

     - Theoretical contributions, architectural invariance, ablation studies, scalability, and uncertainty quantification were acknowledged by the reviewer as having been addressed through the rebuttal and additional experiments.

     - Remaining concerns raised earlier were largely clarified through follow up author responses.

Reviewer 9Cj7

- Concerns that are potentially addressed:

     - Hyperparameter specification and ablation studies were added.

- Concerns that might not be addressed:

    - Thin theory: While the authors strengthened the conceptual grounding by connecting DVA to PFN localization theory and kernel like behavior, the theory remains largely explanatory. It does not provide stronger guarantees such as convergence rates or error bounds, and it is unclear whether this fully satisfies the reviewer’s concern.

    - Scalability: The response frames scalability as an open problem, but does not provide concrete computational complexity analysis of the current method.

    - Baseline concerns: The empirical evaluation remains largely confined to PFN variants and exact GP baselines. Comparisons to other neural surrogate models for high dimensional physical systems are missing, leaving the baseline set narrow and self referential.

Reviewer KGsD

- Concerns that are potentially addressed:

    - Handling of complex priors was clarified.

- Concerns that might not be addressed:

    - Real world tabular data: Although the reviewer confidence is low, the concern appears valid. The paper’s positioning remains ambiguous. If the contribution is intended as a general improvement to PFNs, evaluation on additional tabular datasets would strengthen the claim. If the focus is high dimensional physical systems, comparisons to other physics surrogate models would be needed.


[AC comments]
Positioning issue: It remains unclear whether the paper should be viewed as a general improvement to Prior Data Fitted Networks, or as a method tailored specifically for high dimensional physical surrogate modeling. If the former, evaluation on established real world PFN or tabular benchmarks would be necessary to justify the generality of the claims. If the latter, comparisons against existing neural surrogate models for physical systems would be required to establish relevance and necessity. Many of the reviewer questions seem to be from this ambiguity in positioning and the lack of evidence answering why this method is needed relative to existing alternatives.

**Reviewer Scores:**

Reviewer YKgM: 6 with confidence 3
Reviewer Bixp: 6 with confidence 3 (from 4 / 3)
Reviewer 9Cj7: 4 with confidence 4
Reviewer KGsD: 2 with confidence 1

Even excluding Reviewer KGsD due to very low confidence, the expected average rating remains around 5.3, which is slightly below the acceptance threshold.

---

### Decision · Program_Chairs · 2026-01-26

Reject